



# A niche comparison of *Emiliania huxleyi* and *Gephyrocapsa oceanica* and potential effects of climate change

Natasha A Gafar[1] and Kai G Schulz[1]

[1]Centre for Coastal Biogeochemistry, School of Environment Science and Engineering, Southern Cross University, Lismore, NSW 2480, Australia

**Correspondence:** Natasha Gafar (n.gafar.10@student.scu.edu.au)

**Abstract.** Coccolithophore responses to changes in carbonate chemistry speciation such as $CO_2$ and $H^+$ are highly modulated by light intensity and temperature. Here we fit an analytical equation, accounting for simultaneous changes in carbonate chemistry speciation, light and temperature, to published and original data for *Emiliania huxleyi*, and compare the projections with those for *Gephyrocapsa oceanica*. Based on our analysis, the two most abundant coccolithophores in today's oceans appear to

be adapted for a similar fundamental light niche but slightly different ones for temperature and $CO_2$, with *E. huxleyi* having a tolerance to lower temperatures and higher $CO_2$ levels than *G. oceanica*. Based on growth rates, a dominance of *E. huxleyi* over *G. oceanica* is projected below temperatures of 22°C at current atmospheric $CO_2$ levels. This is similar to a global surface sediment compilation of *E. huxleyi* and *G. oceanica* coccolith abundances suggesting temperature dependent dominance shifts. For a future RCP 8.5 climate change scenario (1000 $\mu$atm $fCO_2$ and + 4.8°C) we project a niche contraction for *G.*

*oceanica* to regions of even higher temperatures. Finally, we compare satellite derived particulate inorganic carbon estimates in the surface ocean with a recently proposed metric for potential coccolithophore success on the community level i.e. the temperature, light and carbonate chemistry dependent $CaCO_3$ production potential (CCPP). Excluding the Antarctic province from the analysis we found a good correlation between CCPP and satellite derived PIC in the other regions with an $R^2$ of 0.73 for Austral winter/Boreal summer and 0.85 for Austral summer/Boreal winter.

**1 Introduction**

Since the Industrial Revolution in the late 18th century, burning of fossil fuels, as well as wide scale deforestation have contributed to significant increases in atmospheric carbon dioxide, $CO_2$ (IPCC, 2013a). Depending upon the decisions in the next few decades, atmospheric $CO_2$ levels are projected to reach between 420 $\mu$atm (RCP2.6 scenario) and 985 $\mu$atm (RCP8.5 scenario) by 2100 (Caldeira and Wickett, 2005; Orr et al., 2005; IPCC, 2013a). To date approximately one third of the anthro-

pogenic carbon emissions have been absorbed by the world's oceans (Sabine et al., 2004). As atmospheric partial pressures of $CO_2$ ($pCO_2$) increase, $CO_2$ concentrations in the surface ocean also increase, resulting in increased bicarbonate and hydrogen ions but also in decreased carbonate ion concentrations and pH (Doney et al., 2009; Schulz et al., 2009). These changes, often termed ocean carbonation and acidification, can have both positive and negative effects for different phytoplankton species and



groups (e.g. Engel et al. 2005; Feng et al. 2010; Moheimani and Borowitzka 2011; Endo et al. 2013; Schulz et al. 2017).

Associated with rising $pCO_2$ is the phenomenon of global warming. Under current scenarios, ocean temperatures are projected to increase from 2.6 to 4.8°C by 2100 (IPCC, 2013b). Warming of the ocean is expected to enhance vertical stratification

of the water column, resulting in a shoaling of the surface mixed layer and increasing overall light availability in the euphotic zone (Bopp et al., 2001; Rost and Riebesell, 2004; Lefebvre et al., 2012). While increased light intensity often accelerates growth in phytoplankton, excessive levels of light can cause damage to the photosynthetic apparatus thus decreasing growth (Powles, 1984; Zondervan et al., 2002).

Coccolithophores play an important role in the marine carbon cycle through the precipitation of calcium carbonate, via calcification and the formation and settling of coccolith aggregates, as well as inorganic carbon fixation by photosynthesis (Rost and Riebesell, 2004; Broecker and Clark, 2009; Poulton et al., 2007, 2010). It is well established that rising $pCO_2$ will have significant effects on coccolithophorid growth, calcification and photosynthetic carbon fixation rates (Riebesell et al., 2000; Bach et al., 2011; Raven and Crawfurd, 2012). Furthermore, it has been shown that the response to rising $pCO_2$ of

both *Gephyrocapsa oceanica* and *Emiliania huxleyi* is strongly influenced by light intensity and temperature (Zondervan et al., 2002; Schneider, 2004; De Bodt et al., 2010; Sett et al., 2014; Zhang et al., 2015). However, to which degree species specific responses may shape individual distribution and abundance in the future ocean is far less clear.

This is because the distribution and abundance of a species is controlled by several factors. Firstly, each species has a specific

range of environmental conditions under which they can successfully grow and reproduce called the fundamental niche. The fundamental niche describes the multi-dimensional combination of environmental conditions, such as temperature, light and $pCO_2$, required for survival of a species assuming no other species are present (Leibold, 1995). However, species do not exist in a vacuum and where the niche of a species overlaps with another species interactions such as competition for resources and predation can occur (Hutchinson, 1957; Leibold, 1995), resulting in the realised niche (Leibold, 1995; Zurell et al., 2016).

Hence it is not only important to determine how environmental change shapes the fundamental niche of individual species, but also consider the impact of niche overlap of different species in shaping the realised niches and hence community composition.

In the present study, we therefore compare species specific sensitivities and responses to combined light, temperature and carbonate chemistry changes of two of the most abundant coccolithophores *Emiliania huxleyi* and *Gephyrocapsa oceanica*.

For that purpose, *E. huxleyi* was grown at twelve $pCO_2$ levels and five light intensities and growth, photosynthetic carbon fixation and calcification rates were measured in response. These data were then combined with a previously published data set on temperature and $CO_2$ interaction (Sett et al., 2014) and fitted to an analytical equation describing the combined effects of changing carbonate chemistry speciation, light and temperature. The resulting projections are then compared to those previously published for *G. oceanica* (Gafar et al., 2018) in an attempt to assess their individual success and potential realised niche

in a changing ocean.



## 2 Methods

### 2.1 Experimental set-up

Mono-specific cultures of the coccolithophore *E. huxleyi* (strain PML B92/11 isolated from Bergen, Norway) were grown in artificial seawater (ASW) at 20°C and a salinity of 35 across a $p\mathrm{CO_2}$ (partial pressure of $\mathrm{CO_2}$) gradient from ∼25-7000 $\mu$atm.

Light intensities were set to 50, 400 and 600 $\mu$mol photons m$^{-2}$s$^{-1}$ of photosynthetically active radiation (PAR) on a 16:8 h light-dark cycle in a Panasonic Versatile Environmental Test Chamber (MLR-352-PE). An additional set of cultures was also incubated at 1200 $\mu$mol photons m$^{-2}$s$^{-1}$ under a Philips SON-T HPS 600W light in a water-bath set to 20°C. Light intensities at each bottle position for all experiments were measured using a LI-193 spherical sensor (LI-COR). Cells were pre-acclimated to experimental conditions for 8-12 generations. To account for differences in growth rate between the extreme high/low $\mathrm{CO_2}$

treatments and the intermediate $\mathrm{CO_2}$ treatments, initial cell densities chosen between 20-80 cells ml$^{-1}$. Treatments were run using a dilute-batch culture setup, mixed daily and harvested before dissolved inorganic carbon (DIC) consumption exceeded 10%.

### 2.2 Media

Artificial seawater (ASW) with a salinity of 35 was prepared according to Kester et al. (1967). ASW was enriched with f/8

trace metals (EDTA bound Fe, Cu, Mo, Zn, Co, Mn) and vitamins (thiamine, biotin, cyanocobalamin) according to Guillard (1975), 64 $\mu$mol kg$^{-1}$ nitrate ($\mathrm{NO_3^-}$), 4 $\mu$mol kg$^{-1}$ phosphate ($\mathrm{PO_4^{3-}}$), 10 nmol kg$^{-1}$ $\mathrm{SeO_2}$ and 1 ml kg$^{-1}$ of coastal seawater (collected at Shelly beach, Ballina, NSW, Australia) to prevent possible limitation by trace elements during culturing which had not been added to the artificial seawater mix. ASW medium was sterile-filtered (0.2 $\mu$m pore size, Whatman$^{TM}$ Polycap 75 AS) directly into autoclaved acclimation (0.5 L) or experimental (2 L) polycarbonate bottles (Nalgene®), leaving a small

head-space for the adjustment of carbonate chemistry conditions.

### 2.3 Carbonate chemistry manipulation, measurements and calculation

Carbonate chemistry, i.e. total alkalinity (TA) and dissolved inorganic carbon (DIC), for each treatment was adjusted through calculated additions of hydrochloric acid (certified 3.571 mol L$^{-1}$ HCl, Merck) and $\mathrm{Na_2CO_3}$ (Sigma-Aldrich, TraceSELECT® quality, dried for 2 hours at 240°C). Samples for TA and DIC measurements were taken at the end of the experiment. TA

samples were filtered through GF/F filters, stored in the dark at 4°C and processed within 7 days (Dickson et al. 2007 SOP 1). TA samples were measured by potentiometric titration using a Metrohm Titrino Plus automatic titrator with 0.05 mol kg$^{-1}$ HCl as the titrant, adjusted to an ionic strength of 0.72 mol kg$^{-1}$ with NaCl (Dickson et al. 2007 SOP 3b).

DIC samples were sterile filtered by gentle pressure filtration with a peristaltic pump (0.2 $\mu$m pore size polycarbonate,

Sartorius) into glass stoppered 100 ml bottles (Schott Duran) with overflow of at least 50% of bottle volume similar to Bockmon and Dickson (2014), sealed without head-space and stored in the dark at 4°C until processing within 7 days. To determine DIC,





ml of sample was analysed on a Marianda AIRICA system by acidification with 10% phosphoric acid to convert all DIC into $CO_2$, followed by extraction with $N_2$ (5.0) and concomitant $CO_2$ analysis with an IR detector (LI-COR LI-7000 $CO_2$/$H_2O$ analyser). Both TA and DIC measurements were calibrated against Certified Reference Materials (batches 139, 141, 150) following Dickson (2010). Initial DIC and TA concentrations were estimated by adding measured total particulate carbon build-up during incubations to measured final DIC, and double the particulate inorganic carbon build-up during incubations to measured final TA concentrations. Carbonate chemistry speciation for each treatment was calculated from mean TA, mean DIC, measured temperature, salinity and $[PO_4^{3-}]$ using the program CO2SYS (Lewis et al., 1998), the dissociation constants for carbonic acid determined by Lueker et al. (2000), $K_S$ for sulphuric acid determined by Dickson et al. (1990) and $K_B$ for boric acid following Uppström (1974).

## 2.4 Particulate organic and inorganic carbon

Sampling started approximately two hours after the onset of the light period and lasted no longer than 3 hours. Duplicate samples for total and particulate organic carbon (TPC and POC) were filtered (-200 mbar) onto GF/F filters (Whatmann, pre-combusted at 500°C for 4 hours) and stored in glass petri-dishes (pre-combusted at 500°C for 4 hours) at -20°C until analysis. POC filters were placed in a desiccator above fuming (37%) HCl for 2 hours to remove all particulate inorganic carbon (PIC). All filters were dried overnight at 60°C, and analysed for carbon content and corresponding isotopic signature according to Sharp (1974) on an elemental analyser (Flash EA, Thermo Fisher) coupled to an isotope ratio mass spectrometer (IRMS, Delta V plus, Thermo Fisher). Particulate inorganic carbon (PIC) was calculated by subtracting measured POC from TPC.

## 2.5 Growth

Cell densities were measured every 3-4 days after the commencement of the experiment using a flow cytometer (Becton Dickinson FACSCalibur) on high flow settings (58 $\mu$l/minute) for two minutes per measurement. Living cells were detected by their red autofluorescence in relation to their orange fluorescence in scatter plots (FL3 vs. FL2). At some extreme $CO_2$ levels there was an initial lag phase and therefore growth rates were calculated from densities only during the exponential part of the growth phase. After disregarding lag phase measurements, the majority of treatments had only two to three data-points in the exponential phase. As a result, specific growth rates were calculated as:

$$\mu = \frac{\ln(C_f) - \ln(C_0)}{d} \tag{1}$$

where $C_f$ represents cell densities at time of sampling, $C_0$ represents cell densities at the beginning of the exponential growth phase, and d is the duration of the exponential phase in days. Calcification and photosynthetic rates were calculated by multiplying cellular PIC and POC quotas with respective growth rates.



## 2.6    Fitting procedure

Coccolithophore metabolic rate (MR) responses of growth, calcification and photosynthetic carbon fixation to combined changes in temperature, light and carbonate chemistry speciation can be described as follows (Gafar et al., 2018).

$$\mathrm{MR}(T,I,S,H) = \frac{k_1 SIT}{k_2 HT + k_3 SHT + k_4 I + k_5 SI + SIT + k_6 SHI^2 T^2}$$ (2)

where, $k_1$ (pg C cell$^{-1}$ day$^{-1}$ or day$^{-1}$), $k_2$ ($\mu$mol photons m$^{-2}$s$^{-1}$), $k_3$ (kg mol$^{-1}$ $\mu$mol photons m$^{-2}$ s$^{-1}$), $k_4$ (mol kg$^{-1}$ °C), $k_5$ (°C), $k_6$ (kg mol$^{-1}$ $\mu$mol photons$^{-1}$ m$^2$s °C$^{-1}$) are fit coefficients, and MR(T,I,S,H) is the metabolic rate of photosynthesis, calcification or growth dependent on temperature (T), light intensity (I), substrate (S = [$CO_2$] + [$HCO_3^-$]) and [$H^+$] (H). Inputs to the equation consisted of calculated $CO_2$, $HCO_3^-$ and $H^+$ (H in total scale) concentrations, as well as measured metabolic rates, and light (I) and temperature (T) levels of all treatments (please see below for information on temperature and
light transforms).

Data from this study (Tables S1, S2) and Sett et al. (2014) were fitted to Eq. (2) using the non-linear regression fit procedure nlinfit in MATLAB (the Mathworks). The reason only these studies were chosen, from the multitude of *E. huxleyi* datasets, is because 1) they use the same strain (PML B92/11), 2) they have the same nutrient conditions and 3) they use the same carbonate
chemistry manipulation methods. Nevertheless, the two chosen studies provided light (six levels) and temperature (three levels) interactions over a broad carbonate chemistry speciation range. It is noted that in both studies the carbonate chemistry system is coupled, meaning that a change in $CO_2$ results in a change in pH. This method reflects the changes in carbonate chemistry speciation due to ongoing ocean acidification (Bach et al., 2011, 2013). However, some studies have examined the effects of decoupled carbonate chemistry where $CO_2$ is changed at a constant pH. This approach is used to tease apart the independent
effects of $H^+$ and $CO_2$ on physiological responses (see Bach et al. 2013). While Eq. (2) can also be used to explain responses under decoupled carbonate chemistry conditions (see Gafar et al. 2018 for details), the fit obtained here is only valid for coupled $CO_2$/pH changes as no data from decoupled experiments (i.e. Bach et al. 2011) has been used. The reason for this being that Bach et al. (2011) does not contain data of temperature, light and carbonate chemistry interactions.

## 2.7    Temperature and light transformations

To reduce skew and to better accommodate certain features (i.e. light and temperature inhibition and limitation) both temperature and light data were transformed. Light data was square root transformed with light (I) = $\sqrt{\mathrm{PFD}}$, where PFD is the photon flux density ($\mu$mol photons m$^{-2}$s$^{-1}$) of an incubation. To accommodate for known temperature inhibition below 2°C and above 30°C (Rhodes et al., 1995; van Rijssel and Gieskes, 2002; Helm et al., 2007; Zhang et al., 2014) at a much narrower experimental range (10-20°C), the upper and lower limits for *E. huxleyi* growth were added into the equation with a general
transform of T = ($T_t$ − 2) × (30 − $T_t$), where $T_t$ is the temperature of an incubation. To accurately express the onset of high temperature inhibition, the transform was further modified with a square root transform to give T = ($T_t$ − 2) × $\sqrt{(30 - T_t)}$.



This transform produces reasonable results when compared to the Eppley temperature envelope curve and the Norberg model (see Gafar et al. 2018).

## 2.8 Physiological rate response parameter estimations to changes in carbonate chemistry, temperature and light

Equation (2) was used to assess the combined effects of carbonate chemistry, temperature and light on growth, calcification and photosynthetic carbon fixation rates, with a focus on general physiological features, such as limitation and inhibition, as well as how much variability could be explained. For growth, photosynthetic carbon fixation and calcification rates optimum $CO_2$ concentrations for maximum production rates ($V_{max}$) and half saturation values were calculated at each experimental light and temperature level. $K_{\frac{1}{2}}$ values consisted of: $K_{\frac{1}{2}CO_2}sat$ which is the $CO_2$ concentration (at certain T and I) at which rates are saturated to half the maximum, and $K_{\frac{1}{2}CO_2}inhib$, which is the $CO_2$ concentration (at certain T and I) at which high proton concentrations reduce physiological rates to half the maximum. Fitting results ($R^2$, fit coefficients, p-values, F-values and degrees of freedom), as well as $V_{max}$, $K_{\frac{1}{2}}$ and $CO_2$ optima are presented in Tables 1, 2 and 3. Species specific differences in response to changing carbonate chemistry, temperature and light were assessed by comparing the above fit to that recently produced for *Gephyrocapsa oceanica* (Gafar et al., 2018).

## 2.9 Niche comparison

To examine the potential of ongoing ocean change to influence realised niches and hence individual success, ranges for light and temperature where both *Emiliania huxleyi* and *Gephyrocapsa oceanica* might be expected to co-exist were selected (i.e. 50-1000 $\mu mol\,photons\,m^{-2}s^{-1}$ and 8-30°C). *E. huxleyi* and *G. oceanica* were chosen for comparison as they are currently the only two species with response data over a range of carbonate chemistry, temperature and light conditions. Growth rates were selected as the point of comparison because they can be used as a measure of relative abundance and therefore dominance of a species, and because growth rates largely control carbon fixation rates. To assess competitive ability, and the potential realised niche, the difference in growth rates between the species was visualised using contour plots.

The effect of temperature on growth rates and hence potential dominance was then compared to phytoplankton community data from global surface sediment samples above the lysocline (McIntyre and Bé, 1967; Chen and Shieh, 1982; Roth and Coulbourn, 1982; Knappertsbusch, 1993; Andruleit and Rogalla, 2002; Boeckel et al., 2006; Fernando et al., 2007; Saavedra-Pellitero et al., 2014). As *E. huxleyi* and *G. oceanica* have similar average numbers of coccoliths per cells, 28 and 21, respectively (Samtleben and Schroder, 1992; Knappertsbusch, 1993; Baumann et al., 2000; Boeckel and Baumann, 2008; Patil et al., 2014), the abundance ratio of *E. huxleyi* to *G. oceanica* coccoliths was here assumed to be a suitable proxy for species dominance. It is noted that *E. huxleyi* has been found to produce excess coccoliths towards the end of blooms when inorganic nutrients become limiting for cellular growth (Balch et al., 1992; Holligan et al., 1993; Paasche, 1998), which would result in an over-estimate of *E. huxleyi* dominance in our study. Nevertheless, given that the coccoliths ratio varies orders of magnitude in modern marine sediments, none of our general conclusions should be affected. Temperature for each sampling site was retrieved from the NOAA 1° resolution annual temperature climatology (Boyer et al., 2013).





### 2.10  Global calcium carbonate production potential

While our fit equation has previously explained variability in lab experiments quite well (Gafar et al., 2018), natural systems
are much more complex, with the interactions of dozens of variables including temperature, light, nutrients, predation and com-
petition all influencing productivity (Behrenfeld, 2014). As such we wanted to examine how our, relatively simple, equation
projections of productivity compared to coccolithophorid productivity patterns observed in natural systems. Productivity can
be defined in a few ways, traditionally, changes in cellular calcification rates, in response to ocean change, have been used as
indicator for the potential success of coccolithophores in the future ocean. However, the exponential nature of phytoplankton
growth amplifies even small differences in cellular growth rates, when applied on the community level. For instance, a phyto-
plankton bloom occurring over one week at a growth rate of $1.0 \, \mathrm{d}^{-1}$ and a starting cell density of 50 cells $\mathrm{ml}^{-1}$ would lead to
a peak density of about 55,000 cells $\mathrm{ml}^{-1}$. This is in stark contrast to conditions where growth is only 10% lower as peak cell
densities, and hence biomass and PIC standing stock, will only be half.

Recently, a new metric was proposed, the $CaCO_3$ production potential (CCPP) which 1) should be a better representation of
potential coccolithophore success on the community level and 2) can be tested against modern observations of surface ocean
$CaCO_3$ distribution. CCPP is defined as the amount of $CaCO_3$ produced within a week by a coccolithophore community
(with a set starting cell count) for a certain environmental condition, calculated from Eq. (2) derived growth rates and inorganic
carbon quotas. Inorganic carbon quotas are calculated as the quotient of calcification and growth rates. As CCPP is calculated
from calcification and growth rates, it accounts for the individual effects of temperature, light and carbonate chemistry on
growth rates and on carbon production. It was for these reasons that CCPP was the metric chosen for comparison.

Provided values for temperature, light, substrate ($CO_2 + HCO_3^-$) and hydrogen ion concentrations (H) for the surface mixed
layer, coccolithophore $CaCO_3$ production potential can be projected for the world oceans. CCPP can then be cautiously evalu-
ated against and compared to satellite derived global particulate inorganic carbon concentration estimates ($PIC_s$). As inorganic
nutrients are a critical factor influencing phytoplankton abundance, and especially bloom formation, in the ocean (Browning
et al., 2017) nitrate concentrations were also included in the analysis (for details see below). As a result, climatological datasets
consisted of, World Ocean Atlas 2013 v2 (WOA) nitrate concentrations at 1° resolution (Boyer et al., 2013); SeaWiFS mixed
layer depth (MLD 2° resolution) from de Boyer Montégut et al. (2004); surface photosynthetically available radiation (PAR
$\mu\mathrm{mol\,photons\,m^{-2}\,s^{-1}}$ 9 km resolution) from the Moderate Resolution Imaging Spectroradiometer (MODIS)-Aqua (NASA
Goddard Space Flight Center, 2014b); diffuse attenuation coefficients at 490nm (9 km resolution) from Pascal (2013); and
NOAA dissolved inorganic carbon, $p\mathrm{CO_2}$, pH (total scale), $[CO_3^{2-}]$, temperature and salinity (4x5° resolution) from Takahashi
et al. (2014). A 9 km resolution climatology for particulate inorganic carbon ($PIC_s$) concentration (mol PIC $\mathrm{m}^{-3}$) was also
retrieved from the Moderate Resolution Imaging Spectroradiometer (MODIS)-Aqua (NASA Goddard Space Flight Center,
2014a). Once acquired, all datasets were interpolated to a 1° resolution.





Hydrogen ion concentrations were calculated as $10^{-pH}$, $CO_2$, after conversion of $pCO_2$ to $fCO_2$ as described in CO2SYS (Lewis et al., 1998), as $[fCO_2]*K0$ (with K0 being the temperature and salinity dependent Henry's constant), $HCO_3^-$ as $[HCO_3^-] = DIC - ([CO_2] + [CO_3^{2-}])$, and substrate (S) as the sum of $CO_2$ and $HCO_3^-$ concentrations. Mean mixed layer nitrate concentrations were calculated by determining concentrations for each depth and averaging from surface to the mixed

layer depth for each grid cell. Mean mixed layer irradiance was calculated in one meter depth increments for each grid cell as

$$I = \sum_{i=1}^{MLD} = \exp^{-k_d(i)} * I_0 \qquad (3)$$

where I is the average PAR ($\mu mol\,photons\,m^{-2}s^{-1}$), $k_d$ is the attenuation coefficient ($m^{-1}$), MLD denotes the mixed layer depth in meters, and $I_0$ is the incident PAR at the surface ($\mu mol\,photons\,m^{-2}s^{-1}$).

Global coverage of oceanic nutrient concentrations are often limited to only a few macro-nutrients (nitrate, silicate, phosphate). However, concentrations of these nutrients are often strongly correlated (e.g. phosphate and nitrate in Boyer et al. 2013). To ensure there was sufficient nutrients to support the level of production estimated by CCPP, we opted to use a single nutrient, i.e. nitrate, in combination with a simple scaling metric. First it was assumed that $CaCO_3$ is produced with a PIC:PON ratio of 6.625 for *E. huxleyi* and 13.25 for *G. oceanica* (based on Redfield proportions and PIC:POC ratios of one and two respec-

tively). Hence, maximum $CaCO_3$ production potential (CCPP$_{max}$) in a grid cell would be 6.625 and 13.25 times the nitrate concentration for *E. huxleyi* and *G. oceanica* respectively. If estimated CCPP for a cell exceeded CCPP$_{max}$, and therefore the nitrate required to produce that much PIC, then it was replaced with the CCPP$_{max}$ value. If CCPP was less than C$_{max}$ then no further changes were applied.

To ensure that mean global CCPP and mean global PIC$_s$ would be of the same magnitude, starting cell counts for CCPP calculations were set at 1 ml$^{-1}$ for *E. huxleyi* alone, 0.25 ml$^{-1}$ for *G. oceanica* alone and 0.25 ml$^{-1}$ for each species when combined. To allow comparison, CCPP and PIC$_s$ were both converted to units of $\mu mol$ PIC L$^{-1}$. All data were then averaged for Austral summer/Boreal winter (December-February) and Austral winter/Boreal summer (June-August). Austral summer/Boreal winter and Austral winter/Boreal summer were chosen as they provide prominent differences between minimum

and maximum PIC, while spring and autumn do not. A direct comparison between PIC$_s$ and CCPP was achieved by splitting results into major ocean biogeographical provinces following Gregg and Casey 2007 with the single change of adjusting the Antarctic and the north ocean regions to start at 45° as in Longhurst 2007 rather than 40° (Figure S1). For each major province, the total amount of PIC$_s$ and CCPP for all comparable grid cells were calculated for Austral summer/Boreal winter and Austral winter/Boreal summer. For comparison, values for each basin and season were then converted into percentages of annual global

(global summer plus global winter) PIC$_s$ or CCPP production. Agreement between the satellite and CCPP estimates was then assessed using a linear correlation.





## 3  Results

The fit equation (Eq. 2) was able to explain up to 85% of the variability in measured metabolic rates of *E. huxleyi* across a broad range of carbonate chemistry (25-4000 $\mu$atm), light (50-1200 $\mu$mol photons m$^{-2}$s$^{-1}$) and temperature (10-20°C) conditions (Table 1).

5  ### 3.1  Responses to changing carbonate chemistry: $CO_2$ and $H^+$

All rates had a similar optimum curve response to the broad changes in carbonate chemistry speciation (Figure 1) regardless of temperature and light intensities. Growth, calcification and photosynthetic carbon fixation rates required similar $CO_2$ concentrations to stimulate rates to half the maximum, $K_{\frac{1}{2}CO_2}sat$ (Table 2, Table 3). Optimum $CO_2$ concentrations for calcification were slightly lower than for photosynthesis or growth (Table 2, Table 3). At $CO_2$ concentrations beyond the optimum, a much higher sensitivity to increasing $[H^+]$, i.e. $K_{\frac{1}{2}CO_2}inhib$ was observed for calcification than for photosynthesis or growth rates (Tables 2, 3 and Figures 1, 2).

### 3.2  Responses to temperature

The effect of temperature on rates was dependent upon $CO_2$, with the greatest effect observed at optimum $CO_2$ concentrations (Figure 1). Increasing temperature increased growth rates up to twofold, photosynthetic rates up to 43% and calcification rates up to 52% (Figure 1, Table 2) under optimal $CO_2$ concentrations. $CO_2$ half saturation concentrations ($K_{\frac{1}{2}CO_2}sat$) were insensitive to temperature (Table 2), while $CO_2$ concentrations for both optimal growth and for inhibition of rates to half the maximum ($K_{\frac{1}{2}CO_2}inhib$) decreased with increasing temperature for all rates (Table 2).

### 3.3  Responses to light

Light intensities affected all physiological rates, with the greatest effect generally being observed at $CO_2$ concentrations at or above the optimum (Figure 2). Between 50 and 1200 $\mu$mol photons m$^{-2}$s$^{-1}$, calcification rates doubled, photosynthetic rates tripled and growth rates increased around 36% (Figure 2, Table 3). Both optimum $CO_2$ and $CO_2$ concentrations at which rates were half saturated ($K_{\frac{1}{2}CO_2}sat$) increased slightly with increasing light intensity (Table 3). $CO_2$ concentrations required to inhibit rates to half of maximum ($K_{\frac{1}{2}CO_2}inhib$) for calcification and photosynthesis increased with increasing light intensity, while those for growth increased from 50-150 $\mu$mol photons m$^{-2}$s$^{-1}$ before decreasing with further increases in light (Table 3).

## 4  Discussion

### 4.1  Responses to changing carbonate chemistry: $CO_2$ and $H^+$

Rates of photosynthesis, calcification and growth in coccolithophores are strongly influenced by $CO_2$ (Bach et al., 2011; Sett et al., 2014; Zhang et al., 2015). Increasing $CO_2$ concentrations resulted in enhanced rates up to an optimum level beyond





which they then declined again. This pattern in growth, photosynthetic carbon fixation and calcification rates has been observed previously for several coccolithophore species (Sett et al., 2014; Bach et al., 2015). The availability of substrate ($CO_2$ and $HCO_3^-$) was suggested as the factor influencing the increase in rates on the left side of the optimum, while the proton concentration ($[H^+]$) was the factor most likely driving declines to the right side of the optimum (Bach et al., 2011, 2015).

Of the two species, *E. huxleyi* has a higher $CO_2$ optimum than *G. oceanica* (Tables 2 and 3, Gafar et al. 2018) for all rates and under most conditions. This could suggest that *E. huxleyi* has a slightly higher substrate requirement than *G. oceanica*. However, considering that *G. oceanica* has both a larger cell size and higher carbon quotas per cell the opposite would be expected (Sett et al., 2014; Bach et al., 2015). An explanation for achieving maximum rates only at higher $CO_2$ concentrations in *E. huxleyi*, in comparison to *G. oceanica* despite a lower inorganic carbon demand, might be a less efficient or capable carbon uptake/ concentrating mechanism. Alternatively, a decreased sensitivity to high $[H^+]$ in *E. huxleyi*, in comparison to *G. oceanica* (see below), would lead to a shift in the optimum towards higher $CO_2$ as well and might be a more likely explanation.

Of the three rates, calcification in *E. huxleyi* had both the lowest $CO_2$ requirement and the highest sensitivity to increasing $[H^+]$ (Tables 3 and 2). This is a pattern previously observed for *G. oceanica* under varying temperature and light conditions (Gafar et al. 2018, See also Table S3). As evidenced by higher $K_{\frac{1}{2}CO_2}inhib$ values for all processes, *E. huxleyi* also appears less sensitive to the inhibiting effects of increasing $[H^+]$ than *G. oceanica* (i.e. $K_{\frac{1}{2}CO_2}inhib$ = 47-250 $\mu mol\,kg^{-1}$ versus 25-99 $\mu mol\,kg^{-1}$ for *G. oceanica* depending on light intensities or $K_{\frac{1}{2}CO_2}inhib$ = 62-250 $\mu mol\,kg^{-1}$ versus 25-130 $\mu mol\,kg^{-1}$ for *G. oceanica* depending on temperature) (Tables 2, 3, S3, Gafar et al. 2018). This also supports earlier results in a model analysis by Bach et al. (2015) where *E. huxleyi* reacted less sensitively to higher $CO_2$ (and $[H^+]$) than *G. oceanica*.

A lower sensitivity of rates to changes in carbonate chemistry speciation, in particular calcification rates, could be explained by the lower degree of calcification in *E. huxleyi* (PIC:POC ratios 0.24-1.38) when compared to *G. oceanica* (PIC:POC ratios 0.82-2.17) (Sett et al., 2014). Higher rates of calcification result in greater production of intracellular $H^+$ ($Ca^{2+} + HCO_3^- \rightleftharpoons CaCO_3 + H^+$), potentially decreasing $[CO_3^{2-}]$ in the coccolith producing vesicle and hence the $CaCO_3$ saturation state (Bach et al., 2015). Furthermore, increased $[H^+]$ has been found to result in declines in $[HCO_3^-]$ uptake, the primary carbon source for calcification (Kottmeier et al., 2016).

## 4.2 Responses to temperature

Temperature was observed to have few modulating effects on $CO_2$ responses in *E. huxleyi*. Changes in temperature produced little (<11 $\mu mol\,kg^{-1}$) change in $CO_2$ optima and substrate saturation ($K_{\frac{1}{2}CO_2}sat$) levels, at least within the measured range (Figure 1, Table 2). Similar results were observed for *G. oceanica* (Gafar et al., 2018). This indicates that while overall rates change, carbon uptake mechanisms appear to scale to maintain internal substrate concentrations and thus cellular requirements regardless of temperature conditions.





In contrast, the inhibition of rates by rising [$H^+$] i.e. $K_{\frac{1}{2}CO_2}inhib$ was more sensitive to temperature. The $CO_2$ concentration at which rates were reduced to half the maximum increased with decreasing temperatures (Table 2). These results were also observed for *G. oceanica* which had a lower sensitivity to increasing [$H^+$] at the lowest tested temperature (Gafar et al., 2018). This also agrees with De Bodt et al. (2010) in which a greater decline in calcification rate was observed with increasing $CO_2$

at 18°C than at 13°C. These results indicate that, at least some, coccolithophores may be less sensitive to high $CO_2$ levels at lower temperatures. As a result, both *G. oceanica* and *E. huxleyi* may become more vulnerable to the negative effects of ocean acidification as ocean temperatures increase due to climate change.

### 4.3    Responses to light

The sensitivity of all rates in *E. huxleyi* to changing carbonate chemistry, in particular increasing [$H^+$], was clearly modulated

by light intensity (Figure 2), agreeing with earlier findings (Zondervan et al., 2002; Feng et al., 2008; Gao et al., 2009; Rokitta and Rost, 2012; Zhang et al., 2015). $CO_2$ half-saturation ($K_{\frac{1}{2}CO_2}sat$) for all rates were insensitive to increasing light intensities (Table S3). This agrees with results for *G. oceanica* which also displayed little change in $CO_2$ half-saturation concentrations with increasing light (Table S3). Increasing light intensity induced increases in $CO_2$ optima in all rates, however these changes were small (<10 $\mu mol\,kg^{-1}$) for calcification and growth rates. This contrasts with *G. oceanica* for which a distinct decrease in

optimal $CO_2$ concentrations for growth rates with increasing light intensities was observed (Table S3). However, *G. oceanica* projections are based on a dataset with only three $CO_2$ concentrations (∼16, 31, 45 $\mu mol\,kg^{-1}$). As such, it is difficult to determine how robust the estimates of $CO_2$ optima and half-saturation requirements may be for this species (Zhang et al., 2015).

In *E. huxleyi* the relationship between $H^+$ sensitivity and light intensity was the same for the three rates. Calcification and photosynthetic carbon fixation and growth rates were most sensitive to $H^+$ at the lowest (50 $\mu mol\,photons\,m^{-2}s^{-1}$) and growth rates were also slightly more sensitive at the highest (1200 $\mu mol\,photons\,m^{-2}s^{-1}$) light intensities (Table 3). This result is in part due to an underestimation of growth rates by the fitting equation under high $CO_2$ conditions at 50 $\mu mol\,photons\,m^{-2}s^{-1}$ light (Figure 2). However, it may be that sub-optimal light intensities add additional stress to the cells resulting in them having

less resources with which to handle the stress of increasing high [$H^+$]. Hence rates are lower, but also appear more sensitive to changing carbonate chemistry. These findings agree with findings by Rokitta and Rost (2012) where a diploid *E. huxleyi* strain became insensitive to the effects of rising $CO_2$ (380 vs. 1000 $\mu atm$) when light intensities were increased from 50 to 300 $\mu mol\,photons\,m^{-2}s^{-1}$. However, this differs to *G. oceanica* which, with rising light intensities, had no change in sensitivity for calcification rates, a decrease in sensitivity for photosynthesis and an increase in sensitivity for growth rates (Table S3).

Again, although this could be indicative for species specific differences in sensitivity, it may also be a result of the low number of $CO_2$ treatments used in the light data of *G. oceanica* (see Zhang et al. 2015).





### 4.4 *E. huxleyi* and *G. oceanica* a niche comparison

In the future ocean $CO_2$, temperature and light availability are all expected to change (Rost and Riebesell, 2004; IPCC, 2013b). Levels of $f CO_2$ are expected to reach as high as 985 $\mu$atm by the end of the century with concomitant rise in global ocean temperature of up to 4.8°C (RCP8.5 scenario IPCC 2013a, b). Light intensities in the surface ocean are also expected to increase as

a result of mixed layer depth shoaling (Rost and Riebesell, 2004). By calculating and comparing growth rates for *E. huxleyi* and *G. oceanica* over a range of environmental conditions, it is possible to differentiate between the fundamental (physiological) niche of a species and its potentially realised niche when in competition with others. For this purpose, light, temperature and $CO_2$ ranges were restricted to those where both species would be expected to co-occur, i.e. 20-1000 $\mu$mol photons m$^{-2}$s$^{-1}$, 8-30°C and 25-4000 $\mu$atm, respectively. The calculated difference in growth rates in response to $CO_2$ and temperature does

not significantly change with light intensity (Figure 3 and 4). It should be noted, however, that light intensity might modify observed growth rate differences for other strains of the same species than used here as they can possess different sensitivities and requirements (i.e. Langer et al. 2009; Müller et al. 2015).

### 4.4.1 Fundamental niche

Experimentally, *E. huxleyi* has been found to grow in a range of ∼6 to 2500 $\mu$mol photons m$^{-2}$s$^{-1}$ with high light resulting

in no inhibition of maximum rates in some strains, and up to 20% reduction in others (Balch et al., 1992; van Bleijswijk et al., 1994; Nielsen, 1995; Nanninga and Tyrrell, 1996; van Rijssel and Gieskes, 2002). In contrast, *G. oceanica* is more sensitive in a similar experimental range of ∼6-2400 $\mu$mol photons m$^{-2}$s$^{-1}$ with maximum rates inhibited by up to 38% at high light intensities (Larsen, 2012). Light intensities below 6 $\mu$mol photons m$^{-2}$s$^{-1}$ for *E. huxleyi* and *G. oceanica* resulted in no growth for both species (van Bleijswijk et al., 1994; van Rijssel and Gieskes, 2002; Larsen, 2012). So, while *G. oceanica* is more

sensitive to high light, the potential upper light limit for growth in both species is beyond naturally occurring maxima. Within this light range both species show a similar increase in projected absolute growth rates of 0-1.57 (d$^{-1}$) for *E. huxleyi* and 0-1.51 (d$^{-1}$) for *G. oceanica* (based on figure 4).

*E. huxleyi* has been successfully cultured at $p CO_2$ levels between ∼20-5600 $\mu$atm, while *G. oceanica* has been successfully

cultured at $p CO_2$ levels of ∼20-3400 $\mu$atm (Sett et al., 2014). Again, the upper tolerance limit for growth in both is not known and well above what is expected for most ocean systems. Responses in projected growth rates with rising $CO_2$ differ between the two species with *G. oceanica* rates dropping to 50% of maximum at $f CO_2$ levels above ∼1760 $\mu$atm while *E. huxleyi* drops to 50% of maximum at ∼5950 $\mu$atm. In terms of temperature *E. huxleyi* has a broader niche of 3-29°C in comparison to *G. oceanica* at 10-32°C. Within this temperature niche both species again show a similar change in absolute growth rates of

0-1.40 (d$^{-1}$) for *G. oceanica* and 0-1.43 (d$^{-1}$) for *E. huxleyi* (based on figure 5).

It should be noted however, that although niche ranges and maximum rates are similar for both species, different requirements ($K\frac{1}{2}sat$) and sensitivities ($K\frac{1}{2}inhib$) will lead to different actual rates at a specific environmental condition. This becomes ev-





ident when examining the temperature, light and $CO_2$ niches to find a combination of conditions at which growth rate for each species is at its maximum. For *E. huxleyi* maximum growth rates of 1.62 ($d^{-1}$) are projected at $\sim$970 $\mu mol\,photons\,m^{-2}s^{-1}$ light, $\sim$640 $\mu atm$ $CO_2$ and 20.2°C. In contrast, the conditions for optimal growth rates of 1.52 ($d^{-1}$) for *G. oceanica* are achieved at $\sim$500 $\mu mol\,photons\,m^{-2}s^{-1}$ light, $\sim$430 $\mu atm$ $CO_2$ and 24.4°C. Differences in sensitivity and therefore perfor-
mance under certain conditions will influence the potentially realised niche of the species. For example, *E. huxleyi* is projected to reach higher growth rates than *G. oceanica* under a broader range of temperature, light and $CO_2$ conditions (Figures 3, 4 and 5), indicating that this species may be more of a generalist.

### 4.4.2   Potentially realised niche

Temperature and $CO_2$ both have substantial effects on the potentially realised niche, of *E. huxleyi* and *G. oceanica* (Figures 4
and 5). In contrast, light intensity has very little effect (Figure 3). *E. huxleyi* appears able to exceed growth rates of *G. oceanica* at temperatures below 22°C under most $CO_2$ and light conditions (Figures 4 and 5). A similar difference in temperature pref-erences has also been observed in New Zealand isolates of *Gephyrocapsa oceanica* and *Emiliania huxleyi* with *G. oceanica* and *E. huxleyi* growing between 10-25°C and 5-25°C at optimum temperatures of 22°C and 20°C, respectively (Rhodes et al., 1995). While these results are based on single strain laboratory experiments, there is evidence that such differences in temper-
ature sensitivity may also hold true in the modern ocean. For example, data gathered from multiple phytoplankton monitoring cruises indicate that while both species are found at higher temperatures, *G. oceanica* largely vanishes from the assemblage at temperatures below 13°C (McIntyre and Bé, 1967; Eynaud et al., 1999; Hagino et al., 2005). However, phytoplankton moni-toring cruises can be seasonally biased and represent a single point in time.

Another way to relate our niche comparison to today's oceans is through surface sediments. Surface sediment samples rep-resent an integrated signal of the composition of a phytoplankton community over time and can therefore be a more suitable proxy of species dominance in a certain location. Global surface sediment data on *G oceanica* and *E. huxleyi* coccolith abun-dance indicates that the dominance of these two species is influenced by temperature, particularly in the Pacific Ocean (Figure 6). Globally the data suggests that dominance switches from *E. huxleyi* to *G. oceanica* at temperatures above 25°C which is
similar to our projections. It is noted, however, that in the Atlantic Ocean there appears to be a warm water *E. huxleyi* strain outcompeting *G. oceanica* at temperatures above 25 degrees. While both species have a similar upper limit to their fundamental thermal niche (i.e. Rhodes et al. 1995), it would appear that the higher minimum temperature of *G. oceanica*, combined with its greater tolerance for high temperatures, restricts its realised niche to the upper end of the temperature range (Figures 4 and 6).

$CO_2$ level also influences the relative growth rates of *E. huxleyi* and *G. oceanica*. Under current day levels of $\sim$400 $\mu atm$, *E. huxleyi* would dominate at temperatures up to 22°C (Figure 5). However, at higher and lower $CO_2$ levels, *E. huxleyi* begins to outgrow *G. oceanica* at progressively higher temperatures. At extreme $CO_2$ levels of 25 and 4000 $\mu atm$ *G. oceanica* is only projected to reach higher growth rates than *E. huxleyi* at temperatures above 29°C (Figure 5). This is also supported by Rhodes et al. (1995) and Bach et al. (2015) which suggest that *G. oceanica* begin to be inhibited at lower $CO_2$ (higher $H^+$)




than *E. huxleyi*. So, while growth rates in both species are negatively affected by increasing $[H^+]$, *G. oceanica* is more sensitive so its rates decrease relative to *E. huxleyi* for the same change in $f\mathrm{CO_2}$. However, this sensitivity is partially mitigated by increasing temperatures. For example, under RCP scenario 8.5 temperature and $\mathrm{CO_2}$ levels are expected to increase up to 4.8 °C and 985 $\mu$atm, respectively. Under higher temperature conditions alone *G. oceanica* would be able to outgrow *E. huxleyi*

under a broader range of $\mathrm{CO_2}$ conditions (Figure 5). Meanwhile, under higher $\mathrm{CO_2}$ conditions alone the thermal niche of *G. oceanica* would decrease with this species being dominated by *E. huxleyi* at temperatures up to 26°C. The combined effect of rising temperature and $\mathrm{CO_2}$ allows *G. oceanica* to outgrow *E. huxleyi* under a broader range of $\mathrm{CO_2}$ conditions but a narrower temperature range. As a result, *G. oceanica*'s niche would be expected to decrease under future ocean conditions.

This comparison only considers *E. huxleyi* and *G. oceanica*. However, coccolithophore communities can be made up of dozens of species (McIntyre and Bé, 1967; Winter and Siesser, 1994), all of which are likely to have different preferences for and sensitivities to changes in $f\mathrm{CO_2}$, temperature and light. Shifts in plankton community structure, as a result of different species and group preferences, in response to environmental change have already been observed in the past (Beaugrand et al., 2013; Rivero-Calle et al., 2015), while simulations also suggest shifts in plankton community under future climate condi-

tions (Dutkiewicz et al., 2015). Species and composition shifts in the coccolithophore communities are likely to alter ocean biogeochemistry with implications for ocean-atmosphere $\mathrm{CO_2}$ partitioning.

### 4.5   Global calcium carbonate production potential

The $\mathrm{CaCO_3}$ production potential (CCPP) is based on cellular $\mathrm{CaCO_3}$ quotas and growth rates calculated for a given set of temperature, light and carbonate chemistry conditions (see section 2.10). Here we test how this measure for productivity com-

pares to estimated surface ocean $\mathrm{CaCO_3}$ content observed by satellite imaging ($\mathrm{PIC_s}$). At this point it is important to remember that CCPP does not account for top-down controls such as grazing or viral attack (Holligan et al., 1993; Wilson et al., 2002; Behrenfeld, 2014), and bottom-up controls such as competition for macro or micro-nutrients (Zondervan, 2007; Browning et al., 2017). Thus, a potential for high $\mathrm{CaCO_3}$ production is not necessarily realised when exposed to different top-down and bottom up pressures.

Calculated CCPP of *E. huxleyi* alone (Figure 7) for the global ocean visually reproduces the mid-latitude production belts, however at lower latitudes than satellite PIC estimates. This agrees with the NEMO and OCCAM models of coccolithophore dominance (Sinha et al., 2010) and the chlorophyll a NASA Ocean Biogeochemical Model (NOBM) model for the Southern hemisphere and central North Atlantic provinces (Gregg and Casey, 2007). CCPP also estimates seasonal changes with higher

productivity during summer in both hemispheres (see figure 7A and D vs. B and E). This pattern is driven mainly by temperature, which influences the latitudinal location of the bands, and light intensity, which influences whether the northern or southern band of productivity is stronger in a season. Nutrients are an essential, and in the ocean often limiting, requirement for biological productivity (Kattner et al., 2004; Browning et al., 2017). As such it would be expected that nutrients should also be strongly influencing seasonal patterns of PIC production. However, with the starting cell concentrations for the CCPP





calculations chosen here, there was sufficient nitrate to support the projected production in most ocean regions (Figure 7C and F). High temperatures drove relatively low productivity in the equatorial regions in agreement with satellite PIC. Similar low levels of coccolithophores are estimated in Sinha et al. (2010) in the equatorial Pacific and Atlantic with the mixed phytoplankton functional group dominating with or without coccolithophores due to low iron and moderate phosphate concentrations and

in Gregg and Casey (2007) for the equatorial Indian and Atlantic provinces. CCPP underestimates production at cold high latitudes, in particular in the Southern Ocean, when compared to the satellite. Similar low levels of coccolithophores have been projected in the Southern Ocean in Gregg and Casey (2007) (very low coccolithophore chlorophyll a), Krumhardt et al. (2017) (growth rates at or close to zero which equates to low to zero CCPP) and Sinha et al. (2010) (high nutrients resulting in coccolithophores being dominated by diatoms). For the Southern Ocean, it has been suggested that satellite PIC concentrations

in subantarctic waters are overestimated by a factor of 2-3 while those in Antarctic waters may be even more so (Holligan et al., 2010; Balch et al., 2011; Trull et al., 2018). The fact that three other global estimates, based on different sets of environmental parameters, all estimate very little productivity in the Southern Ocean seems to support this theory. However, there are also specifically cold adapted strains of *Emiliania huxleyi* found at high latitudes which at least partially could explain discrepancies between the mentioned model projections and satellite derived PIC concentrations (see also below).

In Austral winter/Boreal summer CCPP (for *E. huxleyi*) and satellite PIC estimates closely match ($R^2$=0.73 F=26.78 p<0.01) with low PIC in the South and central South provinces, very low PIC in the equatorial, North Indian and Antarctic provinces and higher PIC in the North central Pacific, North Pacific and North Atlantic provinces (Figure 8A). In Austral summer/Boreal winter CCPP (for *E. huxleyi*) and satellite PIC estimates in individual ocean provinces are also generally of overall good

agreement ($R^2$=0.85 F=50.01 p<0.01). Both CCPP and satellite PIC estimates for Austral summer/Boreal winter are low in all equatorial and North ocean provinces with slightly higher CCPP and satellite PIC production for the North central provinces and higher production in the South and South central provinces (Figure 8B).

Despite having similar PIC patterns, overall PIC estimates can differ significantly between CCPP and PIC$_s$ in some provinces.

These provinces can be divided into two groups characterized by either greater or lesser PIC estimates than those observed by satellite (Figure 8). The mid-latitude provinces of central South and central North Pacific and Atlantic and central South Indic in the summer season belong to the former, with higher CCPP than PIC$_s$. Recently, low phytoplankton biomass in these subtropical gyre systems have been hypothesized to be the result of strong grazing pressure despite high cellular growth rates (Behrenfeld, 2014), lending an explanation of why CCPP is higher than satellite PIC standing stocks. The lower PIC standing

stocks estimated from the satellite could also be the result of other phytoplankton functional groups, such as diatoms, taking a comparatively bigger nutrient share (Iglesias-Rodríguez et al., 2002) thereby leaving less for PIC production by coccolithophores.

In contrast, in Austral summer/Boreal winter in the Antarctic and Austral winter/Boreal summer in the North Pacific, CCPP

is smaller than satellite PIC estimates (Figure 8). *E. huxleyi*, which our projections are based off, has been found to dominate

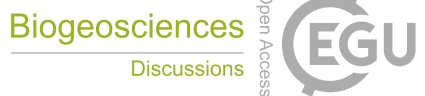


assemblages in polar areas, particularly in the southern hemisphere (Okada and Honjo, 1973; Gravalosa et al., 2008; Mohan et al., 2008; Charalampopoulou, 2011). The strains of *E. huxleyi* found here are special cold-adapted ones which can survive at temperatures as low as -1.7°C in the Antarctic (Cubillos et al., 2007) and -0.9°C in the Arctic (Charalampopoulou, 2011)). As our CCPP is based on a temperate coccolithophore strain, lacking the cold adapted ones, our projections underestimate

coccolithophore productivity in these areas. Additionally, differences in CCPP and satellite PIC in the Southern Ocean may also be connected to satellite overestimation of PIC at high southern latitudes (see above).

Comparing satellite PIC and CCPP in different oceanic provinces (Figure S1) *E. huxleyi* alone provided the greatest agreement between both. The addition of *G. oceanica* to CCPP calculations negatively affected correlations with satellite PIC. This

is counter-intuitive as one would expect increasing correlation of CCPP with satellite PIC as more species are used for the projection of the former. Indeed, estimates based on a combination of *E. huxleyi* and *G. oceanica* in Austral summer/Boreal winter were similar to those for *E. huxleyi* alone. However, in Austral winter/Boreal summer estimates based on a combination of *E. huxleyi* and *G. oceanica* resulted in much lower agreement between CCPP and satellite PIC when compared to *E. huxleyi* alone. This difference is driven by greatly increased CCPP estimates in the central North Pacific and Atlantic, combined with

greatly decreased CCPP estimates in the North Pacific and Atlantic, relative to the *E. huxleyi* alone fit. Being a warm adapted species including *G. oceanica* would result in more productivity in the sub-tropical zones. However, these zones are also regions of potentially significant top-down control (see above for details). Meanwhile the North Pacific and Atlantic are likely dominated by cold-adapted species (see above for details), so including the warm-adapted *G. oceanica* in CCPP calculations would further reduce estimates in these regions. As a result, the inclusion of *G. oceanica* does not assist in making global

estimates of coccolithophore PIC production.

## 5  Conclusions

Our analysis of the projected combination of increased temperature and $CO_2$ on potential success, in terms of growth rates, suggests that *E. huxleyi* will benefit over *G. oceanica*. Due to a greater sensitivity to $CO_2$, *G. oceanica*'s niche will likely contract to regions of higher temperature under future ocean conditions. In general, changes in community composition can

influence community level carbon production and sequestration by coccolithophores. Such changes could have significant implications for climate feedback mechanisms, one being the effects on the relative strength of the organic and inorganic carbon pumps, especially in coccolithophore dominated ecosystems. Temperature and light were found to be important factors driving projections of $CaCO_3$ production potential (CCPP) on a global scale. Comparison of satellite derived inorganic carbon versus estimated inorganic carbon suggests that *E. huxleyi* CCPP is a good proxy for coccolithophore community production in most

biogeographical provinces. However, results indicate that data on the responses of polar species and strains, to environmental change, may be required to improve estimates in the high-latitudes, while the effects of top-down controls might be needed to improve estimates in the mid-latitudes.

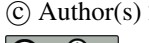



*Data availability.* All data used for the calculation of model fits and coefficients for *Emiliania huxleyi* can be found in the supplementary material for this paper. Fit coefficients used for calculation of *Gephyrocapsa oceanica* niches can be found in Gafar et al. (2018) (DOI: 10.3389/fmars.2017.00433). Third party data sets used for calculation of global calcium carbonate production potential are detailed in Sect. 4.5

5     .

*Author contributions.* Conceived and designed the experiments: KS NG.

Performed the experiments: NG.

Analysed the data: NG KS.

Wrote the paper: NG KS.

10  *Competing interests.* The authors declare that they have no conflict of interest.

*Acknowledgements.* This study was funded by the Australian Research Council (ARC) FT120100384 awarded to KGS and DP150102092 awarded to KGS. We also thank Dr. Matheus Carvalho for analysing particulate carbon samples.



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

**Tables and Figures**





**Table 1.** Fit coefficients ($k_1$ to $k_6$), $R^2$, F-values, degrees of freedom and p-values obtained for calcification (pg C cell$^{-1}$ d$^{-1}$), photosynthetic carbon fixation (pg C cell$^{-1}$ d$^{-1}$) and growth rates (d$^{-1}$) from Eq. (2) fitted to data from this study and Sett et al. (2014). For calcification and photosynthetic carbon fixation rates the unit for $v$ = pg C cell$^{-1}$ day$^{-1}$ while for growth rates the unit for $v$ = day$^{-1}$.

|  | Calcification | Photosynthesis | Growth |
|---|---|---|---|
| $k_1$ (pg C cell$^{-1}$ day$^{-1}$ or day$^{-1}$) | -11.98 | -17.68 | -0.71 |
| $k_2$ ($\mu$mol photons m$^{-2}$ s$^{-1}$) | -1.75E+06 | -4.63E+06 | -9.34E+05 |
| $k_3$ (kg mol$^{-1}$ $\mu$mol photons m$^{-2}$ s$^{-1}$) | 6.43E+07 | 1.39E+09 | 3.10E+08 |
| $k_4$ (mol kg$^{-1}$ °C) | -0.22 | -0.23 | -7.28E-02 |
| $k_5$ (°C) | 28.14 | 26.72 | -38.72 |
| $k_6$ (kg mol$^{-1}$ $\mu$mol photons$^{-1}$ m$^2$ s °C$^{-1}$) | -3.09E+03 | 4.40E+03 | -2.70E+03 |
| $R^2$ (p-value) | 0.7957 (<0.001) | 0.7302 (<0.001) | 0.8460 (<0.001) |
| F-value (degrees of freedom) | 389.51 (100) | 273.52 (100) | 552.74 (100) |



**Table 2.** Optimum $CO_2$ concentrations, $CO_2$ $K_{\frac{1}{2}}$ concentrations and maximum rates ($V_{max}$) at 10, 15 and 20°C from Eq. (2) fit to: $CO_2$-light data at 20°C in this paper and *E. huxleyi* $CO_2$ data from Sett et al. (2014) at 10°C, 15°C and 20°C and 150 $\mu mol\,photons\,m^{-2}\,s^{-1}$ light intensity. Note that the $CO_2$ working range for the equation for this species was 0-250 $\mu mol\,kg^{-1}$. Values exceeding this range were reported as >250 $\mu mol\,kg^{-1}$.

| $CO_2$ | 10°C | 15°C | 20°C |
|---|---|---|---|
| **$CO_2$ optima ($\mu$mol kg$^{-1}$)** | | | |
| Calcification | 16.94 | 12.91 | 11.50 |
| Photosynthesis | 20.34 | 15.42 | 13.91 |
| Growth rate | 29.06 | 20.78 | 18.36 |
| **$V_{max}$** | | | |
| Calcification (pg C cell$^{-1}$ d$^{-1}$) | 6.37 | 8.94 | 9.69 |
| Photosynthesis (pg C cell$^{-1}$ d$^{-1}$) | 8.55 | 11.52 | 12.22 |
| Growth rate (d$^{-1}$) | 0.59 | 1.08 | 1.38 |
| **$K_{\frac{1}{2}\,CO_2}$inhib $\mu$mol kg$^{-1}$** | | | |
| Calcification | 118.47 | 75.04 | 62.94 |
| Photosynthesis | >250 | 119.54 | 100.51 |
| Growth rate | >250 | >250 | 192.74 |
| **$K_{\frac{1}{2}\,CO_2}$sat $\mu$mol kg$^{-1}$** | | | |
| Calcification | 1.66 | 1.56 | 1.48 |
| Photosynthesis | 1.65 | 1.50 | 1.42 |
| Growth rate | 0.85 | 1.19 | 1.40 |





**Table 3.** Optimum $CO_2$ concentrations, $CO_2$ $K\frac{1}{2}$ concentrations and maximum rates ($V_{max}$) at 50-1200 $\mu mol\,photons\,m^{-2}s^{-1}$ from Eq. (2) fit to: $CO_2$ data at 50, 400, 600 and 1200 $\mu mol\,photons\,m^{-2}s^{-1}$ and 20°C in this paper and *E. huxleyi* $CO_2$ data from Sett et al. (2014) at 150 $\mu mol\,photons\,m^{-2}s^{-1}$ light intensity and 10°C, 15°C and 20°C. Note that the $CO_2$ working range for the equation for this species was 0-250 $\mu mol\,kg^{-1}$. Values exceeding this range were reported as >250 $\mu mol\,kg^{-1}$.

| $CO_2$ | 50 PAR | 150 PAR | 400 PAR | 600 PAR | 1200 PAR |
|---|---|---|---|---|---|
| **$CO_2$ optima ($\mu$mol kg$^{-1}$)** | | | | | |
| Calcification | 8.39 | 11.67 | 15.21 | 16.75 | 19.14 |
| Photosynthesis | 9.92 | 14.47 | 21.44 | 26.47 | 52.12 |
| Growth rate | 14.97 | 19.1 | 21.26 | 21.32 | 20.23 |
| **$V_{max}$** | | | | | |
| Calcification (pg C cell$^{-1}$ d$^{-1}$) | 7.64 | 10.05 | 12.47 | 13.48 | 15.04 |
| Photosynthesis (pg C cell$^{-1}$ d$^{-1}$) | 9.16 | 12.78 | 17.27 | 19.82 | 27.24 |
| Growth rate (d$^{-1}$) | 1.19 | 1.43 | 1.58 | 1.61 | 1.62 |
| **$K\frac{1}{2}_{CO_2}$inhib $\mu$mol kg$^{-1}$** | | | | | |
| Calcification | 47.38 | 63.01 | 80.19 | 87.68 | 99.10 |
| Photosynthesis | 73.04 | 104.90 | 182.32 | >250 | >250 |
| Growth rate | 157.71 | 208.62 | 206.04 | 192.60 | 163.64 |
| **$K\frac{1}{2}_{CO_2}$sat $\mu$mol kg$^{-1}$** | | | | | |
| Calcification | 1.00 | 1.53 | 2.13 | 2.39 | 2.81 |
| Photosynthesis | 0.90 | 1.49 | 2.38 | 2.96 | 4.99 |
| Growth rate | 1.08 | 1.46 | 1.69 | 1.73 | 1.72 |





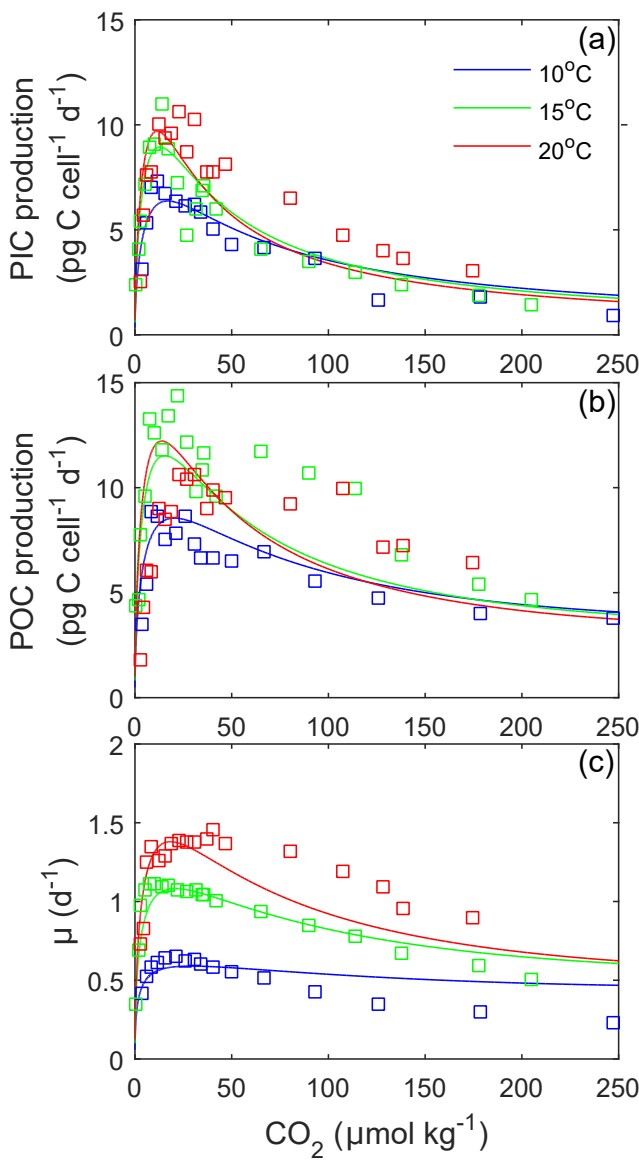

**Figure 1.** (**A**) Fitted particulate inorganic carbon (PIC), (**B**) particulate organic carbon (POC) production, and (**C**) growth rates (solid lines) in response to changes in carbonate chemistry at $10°C$, $15°C$ and $20°C$ using Eq. (2) and fit coefficients from table 1. Symbols represent rate measurements from Sett et al. (2014) at $10°C$, $15°C$ and $20°C$ and $150\,\mu\mathrm{mol\,photons\,m^{-2}s^{-1}}$.





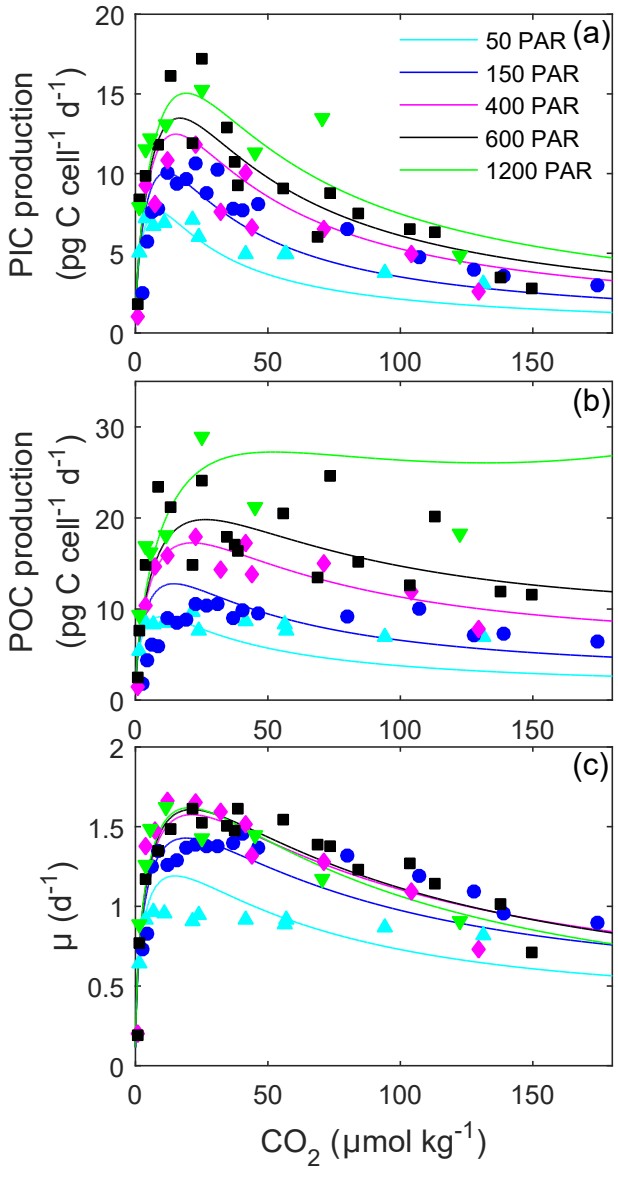

**Figure 2.** Fitted (solid lines) and measured (symbols) (**A**) particulate inorganic carbon (PIC) and (**B**) particulate organic carbon (POC) production, and (**C**) growth rates in response to changes in $CO_2$ concentration at six different light intensities using Eq. (2) and fit coefficients from table 1. Symbols represent rate measurements from this paper at a constant temperature (20°C) and 50, 150, 400, 600 and 1200 $\mu\mathrm{mol\,photons\,m^{-2}s^{-1}}$.





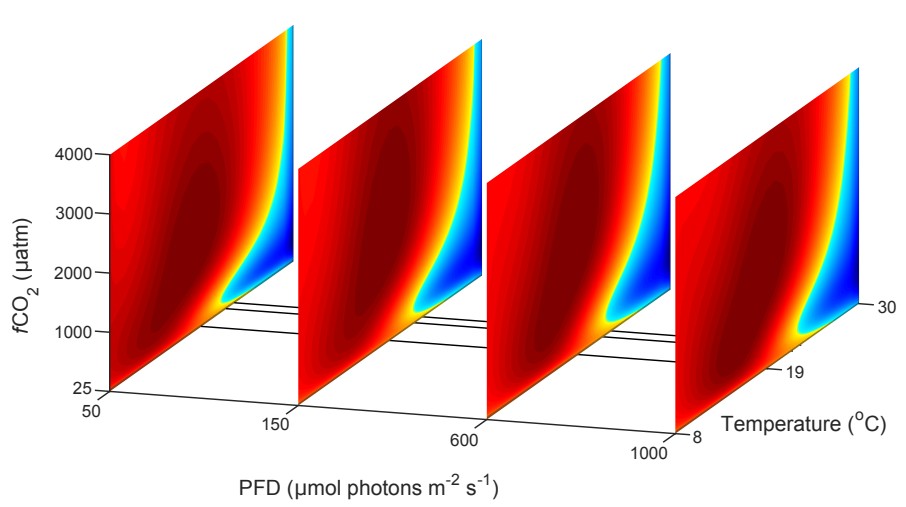

**Figure 3.** Predicted difference in growth rates between *Emiliania huxleyi* and *Gephyrocapsa oceanica* across a temperature range of 8-30°C and a $f\mathrm{CO_2}$ range of 25-4000 $\mu$atm at 50, 150, 600 and 1000 $\mu$mol photons m$^{-2}$s$^{-1}$ of PAR based on Eq. (2). Note the response to varying $\mathrm{CO_2}$ or temperature is not significantly influenced by light intensity.





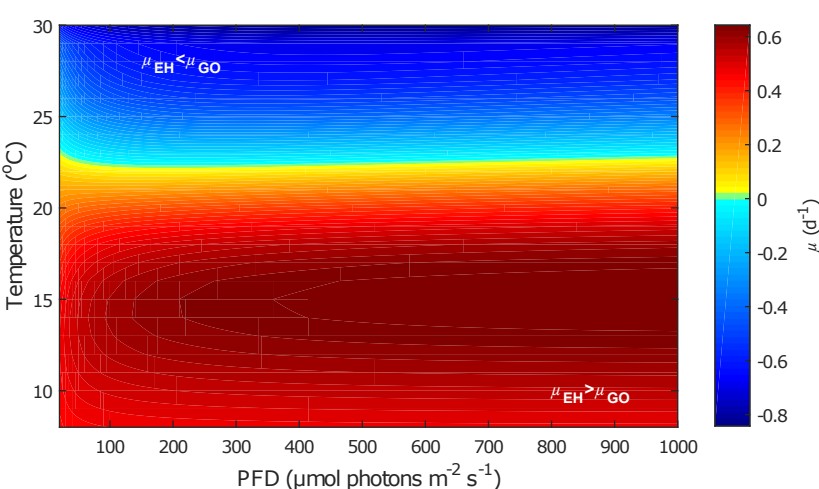

**Figure 4.** Predicted difference in growth rates between *Gephyrocapsa oceanica* and *Emiliania huxleyi* across a light range of 50-1000 $\mu\mathrm{mol\,photons\,m^{-2}s^{-1}}$ and a temperature range of 8-30°C at 400 $\mu\mathrm{atm}\,f\mathrm{CO_2}$. based on Eq. (2).





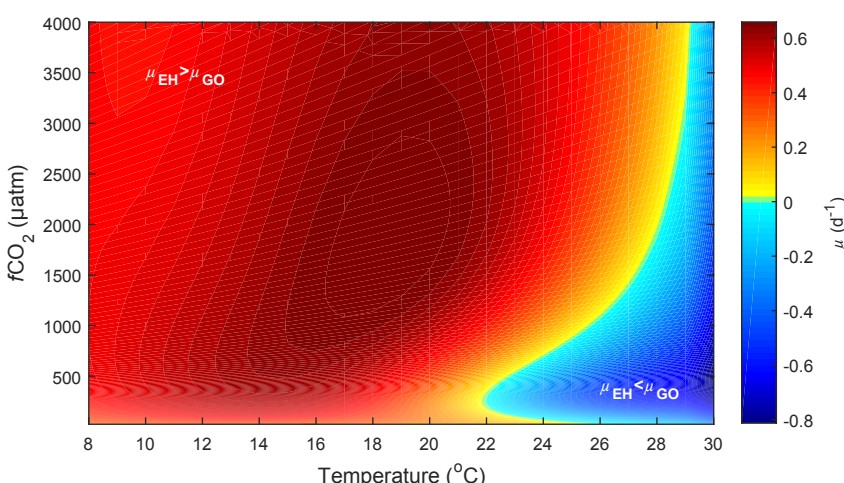

**Figure 5.** Predicted difference in growth rates between *Emiliania huxleyi* and *Gephyrocapsa oceanica* across a temperature range of 8-30°C and a $f\mathrm{CO_2}$ range of 25-4000 $\mu$atm at 150 $\mu$mol photons m$^{-2}$s$^{-1}$ of light based on Eq. (2).





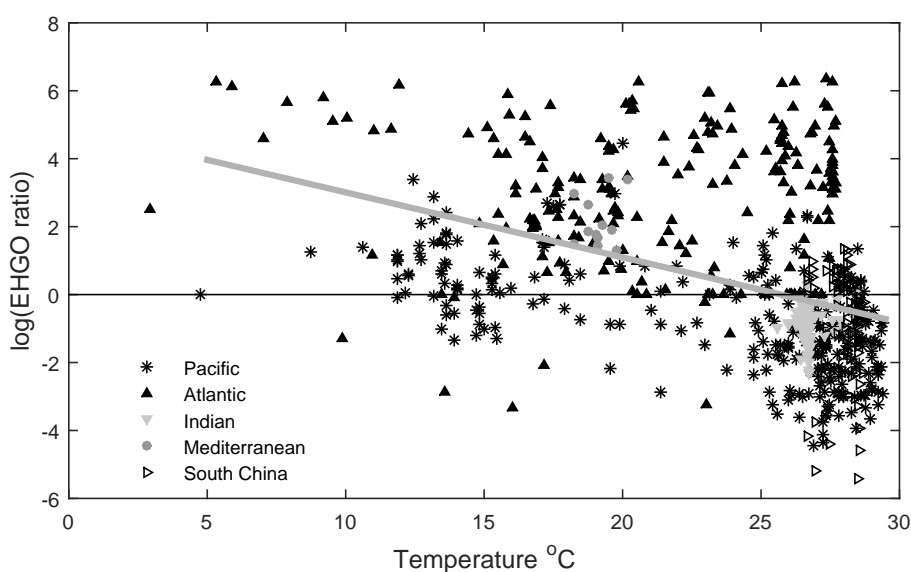

**Figure 6.** Log ratio of *E. huxleyi* to *G. oceanica* coccoliths versus temperature in the global oceans. Symbols and colours represent different ocean basins. The line at zero indicates a shift in dominance from *E. huxleyi* (>0) to *G. oceanica* (<0). The grey line represents a linear regression through the entire dataset with $p < 0.05$ and F of 156.05. For details see Sect. 2.9.



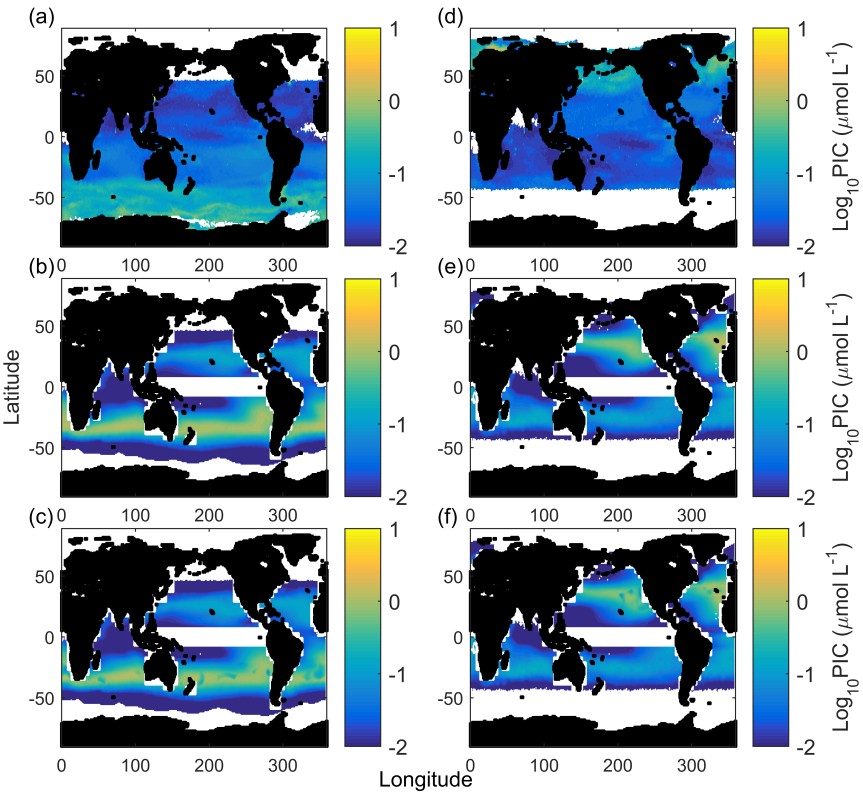

**Figure 7.** Austral summer/Boreal winter (**A**) and Austral winter/Boreal summer (**D**) satellite measured particulate inorganic carbon. Austral summer/Boreal winter (**B**) and Austral winter/Boreal summer (**E**) CCPP estimates accounting for carbonate chemistry (substrate and hydrogen ion concentrations), light intensity and temperature. Note the strong bands of CCPP at the mid-latitudes. Austral summer/Boreal winter (**C**) and Austral winter/Boreal summer (**F**) CCPP estimates accounting for carbonate chemistry (substrate and hydrogen ion concentrations), light intensity and temperature and nitrate concentrations (nutrient proxy).





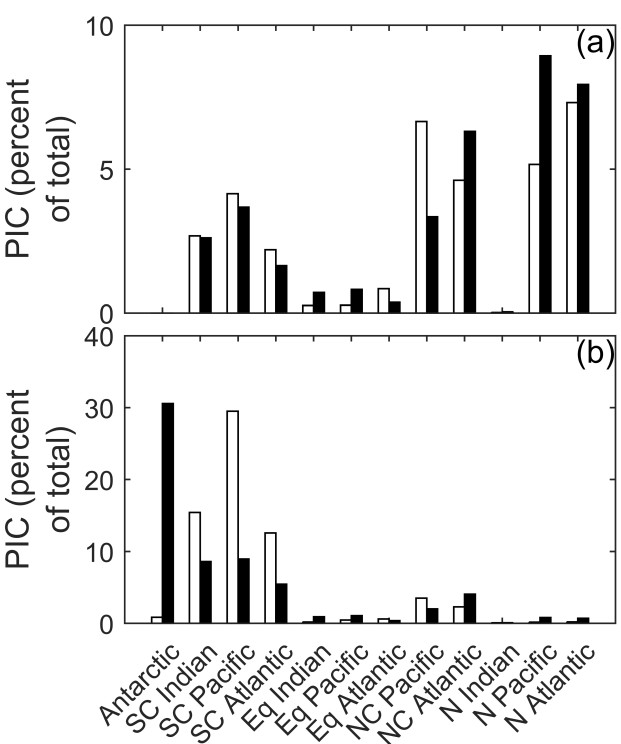

**Figure 8.** Satellite derived particulate inorganic carbon (black bars) and CCPP (white bars) estimates for major ocean biogeographical provinces (see figure S1 for details) as percentages of total production in (**A**) Austral winter/Boreal summer and (**B**) Austral summer/Boreal winter.