# Peer review of "A three-dimensional niche comparison of *Emiliania huxleyi* and Gephyrocapsa oceanica: Reconciling observations with projections"

_Biogeosciences, 2018_

## Short Comment (SC1) · 1 Mar 2018

Dear colleagues

You present a very interesting and useful peace of work. You selected the two species you refer as the most common. Emiliania huxleyi [sorry don't know how to put italics in this text] (Eh) is unquestionably the currently dominating species in oceanic niches. Gephyrocapsa oceanica (Go) is for sure the most abundant but in neritic domain (at least in my area, not sure about Australia), not exactly the most common in the overall oceans. In addition, from a paleoecological point of view, records of Eh are always compared to another small placolith species (small Gephyrocapsids; sG), not to Go,

both in terms of relative and absolute abundances. I understand that Eh and Go are among those coccolithophores that better perform in cultures but shouldn't we compare Eh against sG instead? What's your opinion?

Best regards, Mario Cachao

—————————————————————

---

## Referee Comment (RC1) · A.J. Poulton (Referee) · 23 Mar 2018

This interesting study by Gafar and Schullz presents a new set of laboratory measurements on the coccolithophore species Emiliania huxleyi and Gephyrocapsa oceanica under a broad spectrum of temperature, light and $CO_2$ levels. The authors use this new data, as well as published data from the same strain of E. huxleyi and G. oceanica, to examine potential differences in the biogeographical ranges of the two species in the present and future ocean. The authors conclude that G. oceanica will suffer a considerable niche (range) contraction under future climate change scenarios, which might be unexpected given it generally favours warmer waters than E. huxleyi. The

conclusions of the study are supported by the data generated and I have only minor comments.

Comment 1. Is strain PML B92/11 (isolated from Bergen, Norway) the best strain of E. huxleyi to present a niche description of 'E. huxleyi'? My query here is whether the authors have considered the potential need to consider multiple strains when trying to describe the fundamental and realised niche of the species E. huxleyi. Though the authors state that cold-water (Southern Ocean) strains need to be considered more, would not a broader study of several strains of E. huxleyi, isolated from various geographical regions, result in a better description of the species as a whole? Related to this is whether the authors have considered examining different (geographically) strains of G. oceanica and whether the limitations of G. oceanica's niche could relate to the limited number of strains available for this species? These comments are not meant to detract from the present study, but rather emphasize the broader context.

Comment 2. The authors use a 'recently proposed metric' for coccolithophore calcification rates (CCPP), but proposed by who? No reference is mentioned in the paper. Could the authors provide more context and information on this new metric?

Specific points

Pg 1, Ln 4 (Ln 29) – Emiliania huxleyi is certainly one of the most abundant species, but not sure if G. oceanica can be classified in the same category. The two are common, though E. huxleyi has such a broad bio-geographical range compared to a narrower one for G. oceanica and generally a tropical range. Maybe relative abundance is not the characteristic to emphasize and either a commonality in many coccolithophore communities or bloom-formation by the two is more relevant (to global PIC production).

Pg 1, Ln 13 – As well as the $R^2$ of the correlation, it would be good to know what the slope of the line looks like and the p-value in the abstract.

Pg 8, Ln 13-15 – Have the authors considered how total cell carbon (PIC+POC) to

[Figure]

PON ratios would influence their data? In many ways, the N requirement of a coccolithophore cell is to produce both the PIC and POC. Also, are the PIC:POC ratios of 1 and 2 for E. huxleyi and G. oceanica, respectively, averages of the values given on Lines 23-24? Some justification for the use of the these values, given the ranges known in the literature, is needed.

Pg 14, Lns 21-23 – Surprised the review article by Monteiro et al. (2016) is not mentioned when considering viral attack and top-down effects as this article concluded that these were key considerations in the ecology of coccolithophores.

Pg 16, Lns 2 and 3 – Rather than citing the PhD thesis of Charalampopoulou (2011), why don't the authors cite the peer-reviewed papers derived from this piece of work that address these points?

Charalampopoulou et al. (2011) Irradiance and pH affect coccolithophore community composition on a transect between the North Sea and the Arctic Ocean. Marine Ecology Progress Series 431, 25-43, doi: 10.3354/meps09140.

Charalampopoulou et al. (2016) Environmental drivers of coccolithophore abundance and calcification across Drake Passage (Southern Ocean). Biogeosciences 13, 5917-5935, doi: 10.5194/bg-13-5917-2016.

Pg 16, Ln 23 – Consider the use of the term 'benefit' in terms of E. huxleyi outcompeting G. oceanica.

Pg 16, Ln 27 – What are coccolithophore dominated ecosystems? Please phrase in a more specific way (e.g. where coccolithophores are abundant enough to potentially influence the air-sea CO2 flux (e.g. coccolithophore blooms) or dominate the deep-sea flux of particulate material (e.g. subtropical gyres). Coccolithophores never dominate ecosystems.

Figures.

Fig 3 - Missing legend that is on Fig 4, consider swapping figures around or reproducing

the legend.

Fig 6 – This appears to be a rather unconvincing relationship. Is it possible to plot the 95% CI limits for the relationship? In addition, is there a sampling depth issue here that results in greater amount of data at high temperatures? That is to say, is the distribution of data related to more shallow tropical sediment samples than deep cold sub-polar sediment samples?

Fig 8 – Would it be more appropriate to plot as scatter plots where each data point is from each province? Maybe this would emphasize better how well the two agree and in which provinces they do not agree?

———————————————————

---

## Referee Comment (RC2) · Anonymous Referee #2 · 16 Apr 2018

General comments

In this paper, the authors present laboratory experiments on E. huxleyi that assess the simultaneous impacts of changing CO2, light and temperature. These are compared to previously published data on G. oceanica. This comparison is used to compare niches between E. huxleyi and G. oceanica under present day and future conditions. Further the authors use a CaCO3 production potential metric to compare to satellite derived PIC. There is some interesting/ valuable data and ideas presented in this work, but the paper needs a better cohesion and purpose behind the way the results were presented.

While most of the work focuses on E. huxleyi, the title of the paper suggests that a comparison with G. oceanica is the main topic of the paper. A summarizing comparison figure and a greater clarity as to which figures/tables are only on E. huxleyi would be helpful. I found myself being confused when looking at the figures, tables, and even parts of the text as to which species (or both) is the focus. Since the CCPP comparison to PIC presented in the paper is based only on E. huxleyi, it's difficult to see where the CCPP-PIC data comparison fits into the story presented here. However, it is an interesting result that the addition of G. oceanica does not invoke a better fit.

I would suggest publication with some moderate revisions to the figures, abstract, and some parts of the text.

Detailed comments

Abstract:

As I understand it, the inhibitory effect of increasing CO2 on G. oceanica is the main reason for this species' projected contraction under a future scenario. This should be emphasized in the abstract. As it is now, the projection of a contracted G. oceanica niche is surprising because it is generally the warmer water adapted species.

Also, since E. huxleyi CCPP shows a better correlation with satellite-derived PIC than when combine with G. oceanica, this should be mentioned in the abstract. Otherwise, given the title of the paper, one assumes that the CCPP estimates are derived from partitioning niches between E. huxleyi and G. oceanica.

Also, maybe a sentence at the beginning of Abstract describing why these two partic-ular species are being compared would be helpful.

Intro:

Page 2, lines 3-9: This paragraph on future changes to the surface ocean environment needs expanding. What happens to nutrient availability with increasing stratification? How could this affect CaCO3 production and growth rate in coccolithophores? How

could increasing CO2 affect growth rate/ calcification of coccolithophores? The impact of increasing light is described but not the other effects of climate change. Increasing temperature would also increase metabolic rates, unless nutrient limitation becomes too strong. Overall this paragraph just needs more development with respect to the effects of anthropogenic climate change on coccolithophore habitat and how each effect could impact growth/calcification.

Page 2 $\sim$ line 18: There needs to be a paragraph with some background about the two species discussed in this paper. Why are you comparing these two particular coccolithophore species? These are the two major bloom forming coccolithophores. It is well known that E. huxleyi is very widespread, but how abundant is G. oceanica? Where does G. oceanic tend to thrive? Also mention that there are several different mophotypes of E. huxleyi and how they might differ. A bit of biogeography background would be helpful. This would then lead into the fundamental vs. realized niche paragraph.

Page 2 lines 28-35: the CCPP- PIC comparison is left out of this paragraph. It would be good to mention this here to indicate how it ties in with the E. hux – G. oceanic niche comparison.

Methods:

Page 3, line 4: Why test such high CO2 values? Are these even realistic? For instance, if end of the century CO2 concentration of 985 $\mu$atm (about 50 $\mu$mol kg-1 aqueous CO2), corresponds to a 4.8 deg C temperature increase, then why go up to 250 $\mu$mol kg-1 CO2? The range of CO2 is therefore bigger than the temperature range in terms of real world conditions. An explanation for this experimental setup would be helpful.

Page 3, section 2.1: The authors need to mention the particular E. hux morphotype being tested (PML B92/11 is morphotype A).

Page 4, line 22: Why would there be a lag phase? It seems the growth rate is calculated correctly (after the lag phase is over), but a quick explanation of why there is a lag phase

at extreme CO2 and whether this is a normal phenomenon in phytoplankton culturing and physiological testing would be helpful.

Page 5, section 2.7: I find this section about the data transformation confusing, particularly about the temperature. Is this just for growth rate? How do the resulting temperature-dependent growth rates compare to other studies on coccolithophores (Fielding 2013, Buitenhuis et al., 2008)?

Page 7, line 4: Unneeded commas before and after "relatively simple" . . . or just rewrite for clarity "As such we wanted to examine how projections of productivity using our relatively simple equation compared to coccolithophorid productivity patterns observed in natural systems"

Page 7, line 13: A citation of the CCPP metric is needed.

Page 8, line 14: Need citation for the PIC:POC ratios used for E. huxelyi and G. oceanica

Page 8, last paragraph: I took me awhile to figure out the CCPP estimates were made in three ways: 1) just E. huxleyi 2) just G. oceania 3) both species combined

Is this correct? Only results for E. huxleyi CCPP was presented so maybe clarify here that only the results with the highest correlation to satellite PIC are shown. It's confusing because there are details described in the previous paragraphs about deriving CCPP for each species but then the results only show E. huxleyi CCPP.

Page 8, lines 26 and 27: Need parentheses around year for citations Gregg and Casey (2007) and Longhurst (2007).

Results:

Page 9, Results section in general: Please specify in the headings that these are only the results for E. huxleyi (not G. oceanica).

Page 9, line 2: Perhaps develop this small section a bit more. Which rate showed the

best fit?

Page 9, line 6: Instead of just saying "all rates", please remind the reader what metabolic rates you are examining and refer to the equation presented in the methods.

Page 9, line 7: It's hard to understand exactly what to look at in Table 2 and 3 to support this sentence (2nd sentence of the paragraph). It seems like $CO_2$ concentrations of K1/2sat range form 0.85 to almost 5 $\mu$mol kg-1 depending on light and temperature...

Page 9, lines 8-10: Mention what are the optimal $CO_2$ concentrations and put this into units of $\mu$atm to make it more relatable to the reader. Are we at the optimum $CO_2$ already for coccolithophores or will it come in the near future? At what $CO_2$ concentrations is K1/2inhib reached? More specifics would give the reader more useful information.

Page 9, line 14/15: What columns in table 2 are the reader supposed to be looking at? Are you referring to the Vmax column?

Page 9, line 18: I had to read this sentence several times before I actually understood it. Would this be a better way to put this?: "$CO_2$ half saturation concentration were insensitive to temperature. However, under increasing temperatures $CO_2$ optima for growth and inhibition occurred at lower $CO_2$ concentrations"

Discussion:

Page 10, line 6: Since this is a major conclusion of the paper, it should be shown directly somehow. All the original G. oceanica data is published elsewhere, so a graphical summary of BOTH the E. hux and G. oceanic data would be helpful. This could be done through line plots comparing the metabolic rates of the two species under varying $CO_2$ or in a bar plot comparing the rates. I just think it's necessary to show a visual comparison of E hux and G oceanica data (or data-derived function) since the title of the paper indicates a comparison.

Page 10, line 30: A change in CO2 optima of 11 $\mu$mol kg-1 is not that small.

Page 11, line 5: Unneeded commas around "at least some"

Page 11, line 15: Again, here is where a comparison figure between E. hux and G oceanica would be helpful.

Page 12, line 3: The range tested in this study is so much higher than even what projected under RCP8.5 at the end of the century. The temperature range tested in this study is much more realistic. How warm would the world be under 5000 $\mu$atm CO2?

Page 12, line 12: I think a major limitation of this study is the focus on just one strain of one morphotype of E. huxleyi. Different E. huxelyi morphotypes show significant genetic and physiological variability (see Read et al., 2013; Langer et al., 2009; Krumhardt et al., 2017). Accounting for these differences could add significant uncertainty to the conclusions. I think that the last sentence of section 4.4 (before section 4.4.1) would fit better in a section on the "Limitations of this study" at the end of the Discussion section, where you describe how E huxleyi strain PML B92/11 is used to be representative of all E. huxleyi for determining niche and projections under future CO2 and warming in this study. This doesn't make the results invalid, but is just a limitation that needs to be made clearer. This would then lead in nicely with the conclusion that more testing with colder water strain/species/morphotypes of E huxleyi is necessary.

Page 12, line 22: Capitalize "Figure"

Page 13, line 7: I think it's well established that E. huxleyi is a generalist, given its widespread distribution from subpolar to tropics.

Page 13, line 9: Unneeded comma after "niche"

Page 13, line 25: Reference needed for this E. huxleyi warm water strain that outcompete G. oceanica at temps > 25C

Page 13, line 32/33: I'm confused by this lower CO2 extreme of 25 $\mu$atm. By Figure 5 it looks like G. oceanica outcompetes E hux at temps > 25C at 25 $\mu$atm CO2.

Page 14, lines 1-3: This is a major finding of this study and should be put in the abstract.

Page 14, line 4/5: The sentence seems like it shouldn't have the "under a broader range of CO2 conditions" part at the end. Under higher temperature alone (holding CO2 at about 400$\mu$atm) G. oceanica outcompetes E hux at temps > 22C. Or perhaps I'm misunderstanding this sentence completely?

Page 14, last paragraph of section 4.4.2: This would be better in a "Limitations of this study" section, as mentioned above.

Page 15, line 12: By "productivity", do you mean calcification?

Page 15, lines 14-22: Would this paragraph better fit in the Results section?

Page 16, lines 8-20: Could it be that E huxleyi CCPP just matches better because it's so much more abundant than G. oceanica?

Page 16, line 14/15: I do not understand this sentence. So the combined CCPP in the North Pacific and Atlantic is greater or less than the E huxleyi CCPP?

Tables:

Tables 2 and 3: Put parentheses around units for K1/2CO2inhib and K1/2CO2sat in tables.

Figures:

Figures 1 and 2: Indicate that this data is just for E. huxleyi in the caption. Also, show relevant CO2 range with a shaded area as in Sett et al., 2014 and indicate average oceanic CO2 concentration at present day.

Figure 3: Each "slice" looks the same.. maybe there's a better way to show differences between light levels or lack thereof? Also I do not understand the colors – add color

legend.

Figure 4: Make the $\mu$EH > $\mu$GO bigger or put it next to the color bar. It's a bit hard to notice and this is critical for understanding the figure.

Figure 5: same suggestion as for Figure 4.

Figure 7: It needs to be mentioned in the caption that these maps are CCPP for E huxleyi only.

Figure 8: Again, this is just CCPP for E huxleyi, right? This should be indicated in the figure caption. Also, a little map of the provinces (like in the supplemental section) would be great next to these bar plots. Having a map next to this data would make the figure much more relatable.

References cited:

Fielding, Samuel R. "Emiliania huxleyi specific growth rate dependence on temperature." Limnology and Oceanography58, no. 2 (2013): 663-666.

Buitenhuis, Erik T., Tanja Pangerc, Daniel J. Franklin, Corinne Le Quéré, and Gill Malin. "Growth rates of six coccolithophorid strains as a function of temperature." Limnology and Oceanography 53, no. 3 (2008): 1181-1185.

Read, Betsy A., Jessica Kegel, Mary J. Klute, Alan Kuo, Stephane C. Lefebvre, Florian Maumus, Christoph Mayer et al. "Pan genome of the phytoplankton Emiliania underpins its global distribution." Nature 499, no. 7457 (2013): 209.

Langer, Gerald, Gernot Nehrke, Ian Probert, J. Ly, and Patrizia Ziveri. "Strain-specific responses of Emiliania huxleyi to changing seawater carbonate chemistry." Biogeosciences 6, no. 11 (2009): 2637-2646.

Krumhardt, Kristen M., Nicole S. Lovenduski, M. Debora Iglesias-Rodriguez, and Joan A. Kleypas. "Coccolithophore growth and calcification in a changing ocean." Progress in Oceanography (2017).

---

## Author Comment (AC1) · 2 May 2018

-Comment 1. Is strain PML B92/11 (isolated from Bergen, Norway) the best strain of E. huxleyi to present a niche description of 'E. huxleyi'? My query here is whether the authors have considered the potential need to consider multiple strains when trying to describe the fundamental and realised niche of the species E. huxleyi. Though the authors state that cold-water (Southern Ocean) strains need to be considered more, would not a broader study of several strains of E. huxleyi, isolated from various geo-graphical regions, result in a better description of the species as a whole? Related to this is whether the authors have considered examining different (geographically) strains

of G. oceanica and whether the limitations of G. oceanica's niche could relate to the limited number of strains available for this species? These comments are not meant to detract from the present study, but rather emphasize the broader context.

We agree that considering multiple strains, from diverse ocean regions, would benefit our study in describing the fundamental and realised niches for a species in more general terms. Nevertheless, despite the fact that our realised niche projections are based on only one strain for each species, they do generally fit to modern day observations. This indicates that the differences in requirements and sensitivities of the two species as described here are large enough to be revealed by choosing only two representatives.

-Comment 2. The authors use a 'recently proposed metric' for coccolithophore calcification rates (CCPP), but proposed by who? No reference is mentioned in the paper. Could the authors provide more context and information on this new metric?

This metric was proposed by us in a recently published paper. We will add the reference for this metric where it is mentioned in the main body of the paper.

-Specific points -Pg 1, Ln 4 (Ln 29) – Emiliania huxleyi is certainly one of the most abundant species, but not sure if G. oceanica can be classified in the same category. The two are common, though E. huxleyi has such a broad bio-geographical range compared to a narrower one for G. oceanica and generally a tropical range. Maybe relative abundance is not the characteristic to emphasize and either a commonality in many coccolithophore communities or bloom-formation by the two is more relevant (to global PIC production).

We agree and would change to: "the two most common bloom-forming species in present day coccolithophore communities appear adapted to…"

-Pg 1, Ln 13 – As well as the R2 of the correlation, it would be good to know what the slope of the line looks like and the p-value in the abstract.

We will add the p-value to the abstract and discuss the slope of the line in the text in more detail. Austral summer p-value= 5.46e-05, slope=0.32, Austral winter p-value= 3.06e-04, slope=1.03.

The reason for the relatively small slope of 0.32 in Austral summer, meaning that we overestimate the total production by a factor of three, are the high values of satellite derived PIC in the Antarctic province (which for several reasons given in the MS were not included in the correlation analysis). To rectify this issue, a simple scaling factor could be introduced.

-Pg 8, Ln 13-15 – Have the authors considered how total cell carbon (PIC+POC) to PON ratios would influence their data? In many ways, the N requirement of a coccolithophore cell is to produce both the PIC and POC. Also, are the PIC:POC ratios of 1 and 2 for E. huxleyi and G. oceanica, respectively, averages of the values given on Lines 23-24? Some justification for the use of the these values, given the ranges known in the literature, is needed.

We have. When calculating maximum supportable carbon production, we first assumed a Redfield ratio of 106:16. This value gave us the maximum POC production from the amount of available nitrate. We then calculated the amount of PIC which would be co-produced, with the POC, based on a mean PIC:POC ratio. So for the calculations both PIC:POC and POC:PON ratios were considered. This will be made more clear in the methods section.

PIC:POC values will be amended and based on average PIC:POC of E. huxleyi and G. oceanica from all treatments between 300-1000 $\mu$atm from Sett et al. 2014, Zhang et al. 2015 and this study. This will be mentioned within the methods section.

-Pg 14, Lns 21-23 – Surprised the review article by Monteiro et al. (2016) is not mentioned when considering viral attack and top-down effects as this article concluded that these were key considerations in the ecology of coccolithophores.
The review article will be added.

-Pg 16, Lns 2 and 3 – Rather than citing the PhD thesis of Charalampopoulou (2011), why don't the authors cite the peer-reviewed papers derived from this piece of work that address these points? Charalampopoulou et al. (2011) Irradiance and pH affect coccolithophore community composition on a transect between the North Sea and the Arctic Ocean. Marine Ecology Progress Series 431, 25-43, doi: 10.3354/meps09140. Charalampopoulou et al. (2016) Environmental drivers of coccolithophore abundance and calcification across Drake Passage (Southern Ocean). Biogeosciences 13, 5917-5935, doi: 10.5194/bg-13-5917-2016.

We will adopt the reviewer's suggestion.

-Pg 16, Ln 23 – Consider the use of the term 'benefit' in terms of E. huxleyi outcompeting G. oceanica.

Instead "E. huxleyi will gain further competitive advantage over G. oceanica.

-Pg 16, Ln 27 – What are coccolithophore dominated ecosystems? Please phrase in a more specific way (e.g. where coccolithophores are abundant enough to potentially influence the air-sea $CO_2$ flux (e.g. coccolithophore blooms) or dominate the deep-sea flux of particulate material (e.g. subtropical gyres). Coccolithophores never dominate ecosystems.

We will change to: "Such changes could have significant implications for climate feedback mechanisms, one being the relative strengths of the organic and inorganic carbon pumps in ecosystems where coccolithophores are abundant enough to significantly impact the air-sea $CO_2$ flux (e.g. coccolithophore blooms) and/or dominate the deep-sea flux of particulate material (e.g. subtropical gyres).

-Figures. -Fig 3 - Missing legend that is on Fig 4, consider swapping figures around or reproducing the legend.

We will reproduce the legend on Figure 3.

-Fig 6 – This appears to be a rather unconvincing relationship. Is it possible to plot the 95% CI limits for the relationship? In addition, is there a sampling depth issue here that results in greater amount of data at high temperatures? That is to say, is the distribution of data related to more shallow tropical sediment samples than deep cold sub-polar sediment samples?

We will add 95% prediction bounds for new observations for the global relationship. The fact that only the Atlantic basin does not entirely follow the trend has been mentioned in the text of the paper as well. The data which does not follow the overall pattern (which now will be marked with a different symbol on the plot) is from the south-equatorial to equatorial zone, taken from a study by Boeckel et al. 2006. In this study it appears that G. oceanica abundance is driven more by increasing nutrient concentrations than by temperature. E. huxleyi seems to also be driven by increasing nutrients but it also dominates more in the colder regions. It seems the upwelling in this region is driving a different relationship between E. huxleyi and G. oceanica than in other areas. We shall try to explain this more clearly within the relevant section of the paper.

There is very little sediment sampling data at high latitudes in general let alone which contains these species. There is a disproportionately large amount of sampling in the warmer tropical areas. Also, we selected only samples which were above the lyso-cline and therefore were not affected by the possible confounding effects of differential dissolution of coccoliths.

-Fig 8 – Would it be more appropriate to plot as scatter plots where each data point is from each province? Maybe this would emphasize better how well the two agree and in which provinces they do not agree?

While using a scatterplot does emphasise that the two do not agree in some provinces, it also makes it more difficult to determine which provinces do and do not agree. For the purpose of clearly comparing each province in each season quickly, and having now included more details and a discussion on the slope of the fits, we feel that the

barplot works best.

---

## Author Comment (AC2) · 2 May 2018

-Abstract: -As I understand it, the inhibitory effect of increasing $CO_2$ on G. oceanica is the main reason for this species' projected contraction under a future scenario. This should be emphasized in the abstract (1). As it is now, the projection of a contracted G. oceanica niche is surprising because it is generally the warmer water adapted species. Also, since E. huxleyi CCPP shows a better correlation with satellite-derived PIC than when combine with G. oceanica, this should be mentioned in the abstract (2). Otherwise, given the title of the paper, one assumes that the CCPP estimates are derived from partitioning niches between E. huxleyi and G. oceanica. Also, maybe a sentence

at the beginning of Abstract describing why these two particular species are being compared would be helpful.

1. We will modify to "For a future RCP 8.5 climate change scenario (1000 $\mu$atm fCO2 and + 4.8C) we project a primarily CO2 driven niche contraction for G. oceanica to regions of even higher temperatures." 2. We will modify to "CCPP estimates were based on E. huxleyi alone as interestingly there was a better correlation with satellite-derived PIC than when in combination with CCPP for G. oceanica. Excluding the Antarctic province from the analysis we found a good correlation between CCPP and satellite derived PIC in the other regions with an R2 of 0.73 for Austral winter/Boreal summer and 0.85 for Austral summer/Boreal winter." 3. We will modify the beginning of the abstract to "Based on our analysis of the two most common coccolithophores in today's ocean..."

-Intro: -Page 2, lines 3-9: This paragraph on future changes to the surface ocean environment needs expanding. What happens to nutrient availability with increasing stratification (1)? How could this affect CaCO3 production and growth rate in coccolithophores (2)? How could increasing CO2 affect growth rate/ calcification of coccolithophores (3)? The impact of increasing light is described but not the other effects of climate change. Increasing temperature would also increase metabolic rates, unless nutrient limitation becomes too strong. Overall this paragraph just needs more development with respect to the effects of anthropogenic climate change on coccolithophore habitat and how each effect could impact growth/calcification.

-The potential effects of CO2 on phytoplankton in general and coccolithophores in particular are already covered in the previous and following paragraph. Nevertheless, we will modify the text to "Depending on emission scenarios, ocean temperatures are projected to increase from 2.6 to 4.8C by 2100 (IPCC, 2013b). In addition, warming of the ocean is expected to enhance vertical stratification of the water column, resulting in a shoaling of the surface mixed layer and increasing overall light and decreasing nutrient availability in the euphotic zone (Bopp et al., 2001; Rost and Riebesell, 2004; Lefeb-

vre et al., 2012). While increased light intensity and temperatures often accelerate growth in phytoplankton, excessive levels of light and temperature can cause damage to the photosynthetic apparatus and reduce effectiveness of enzymes thus decreasing growth (Powles, 1984; Rhodes et al. 1995; Crafts-Brandner 2000; Zondervan et al., 2002; Helm et al., 2007; Reviewed in Pörtner and Farrell, 2008). Meanwhile, reduced nutrient availability could diminish overall productivity."

-Page 2âĹijline 18: There needs to be a paragraph with some background about the two species discussed in this paper. Why are you comparing these two particular coccolithophore species? These are the two major bloom forming coccolithophores. It is well known that E. huxleyi is very widespread, but how abundant is G. oceanica? Where does G. oceanic tend to thrive? Also mention that there are several different mophotypes of E. huxleyi and how they might differ. A bit of biogeography background would be helpful. This would then lead into the fundamental vs. realized niche paragraph.

-We will add the following text in line 14 after "(Rost and Riebesell, 2004; Broecker and Clark, 2009; Poulton et al., 2007, 2010)." The coccolithophores Emiliania huxleyi and Gephyrocapsa oceanica are considered the most common species in present day coccolithophore communities. E. huxleyi is a ubiquitous coccolithophore species having being observed from polar to equatorial regions, nutrient poor ocean gyres to nutrient rich upwelling systems, and from the bright sea surface down to 200m depth (McIntyre & Be 1967; Winter et al. 1994; Hagino & Okada 2006; Boeckel & Baumann 2008; Mohan et al. 2008; Henderiks et al. 2012). The wide tolerance of E. huxleyi to different environmental conditions is believed to be, at least partially, explained by the existence of a number of environmentally selected ecotypes and morphotypes within the species (Paasche 2001; Cook et al. 2011). G. oceanica is also found in most oceanographic regions (McIntyre and Be 1967; Okada and Honjo 1975; Roth and Coulbourn 1982; Knappertsbusch et al. 1993; Eynaud et al., 1999; Andruleit et al. 2003; Saaveda-Pellitero et al. 2010), however with a tendency towards warmer waters with very few

specimens observed below 13oC (McIntyre and Bé, 1967; Eynaud et al., 1999; Hagino et al., 2005).

-Page 2 lines 28-35: the CCPP- PIC comparison is left out of this paragraph. It would be good to mention this here to indicate how it ties in with the E. hux – G. oceanic niche comparison.

We will add the following to the end of the paragraph "Finally, we compare satellite derived particulate inorganic carbon estimates with a recently proposed metric for coccolithophore success on the community level (Gafar et al. 2018), i.e. the temperature, light and carbonate chemistry speciation dependent calcium carbonate potential.

-Methods: -Page 3, line 4: Why test such high CO2 values? Are these even realistic? For instance, if end of the century CO2 concentration of $985\mu$atm (about $50\mu$mol kg-1 aqueousCO2), corresponds to a 4.8 deg C temperature increase, then why go up to $250\mu$mol kg-1 CO2? The range of CO2 is therefore bigger than the temperature range in terms of real world conditions. An explanation for this experimental setup would be helpful.

Fitting non-linear responses of multiple stressors to data requires a broad range of environmental conditions, as otherwise the shaping factors of limitation and inhibition are lost (absent from data while present in model equation). With this broader range we also have the added benefit for identifying tipping points and changes in sensitivities to CO2 with changing light and temperature.

We will make our rational more clear in the methods section by adding the following to Page 3 line 3: "To accurately identify optimal conditions, tipping points and sensitivities of rates in response to changing CO2, light and temperature, a broad range of experimental conditions were required. Mono-specific. . . . . .."

-Page 3, section 2.1: The authors need to mention the particular E. hux morphotype being tested (PML B92/11 is morphotype A).

We will add the requested information.

-Page 4, line 22: Why would there be a lag phase? It seems the growth rate is calculated correctly (after the lag phase is over), but a quick explanation of why there is a lag phase at extreme CO2 and whether this is a normal phenomenon in phytoplankton culturing and physiological testing would be helpful.

-At both, the extreme low and high CO2 treatments, carbonate chemistry at the end of the pre-incubation phase can significantly deviate from initial and hence experimental treatment conditions due to enhanced air/water CO2 gas exchange during regular cell abundance monitoring. This in turn can induce a lag phase at the beginning of experimental conditions as observed here. We will add this information to the method's section.

-Page 5, section 2.7: I find this section about the data transformation confusing, particularly about the temperature. Is this just for growth rate? How do the resulting temperature-dependent growth rates compare to other studies on coccolithophores (Fielding 2013, Buitenhuis et al., 2008)?

This transform is applied to all rates to reduce skew and are common practice in multivariate fitting procedures.

As mentioned within section 2.7, this temperature transform compares well to other temperature dependant growth rate equations such as the single species responses to the Eppley temperature envelope curve and the Norberg model. Our temperature-dependant growth rate estimates show a similar response to the optimal growth function in Buitenhuis et al. 2008 and the Flinn equation in Fielding et al. 2013. The power function in Fielding et al. 2013 also follows a similar pattern, of growth rate increase with rising temperature, as our transform but lacks a term to inhibit rates as temperatures rise above optimum.

However, our temperature transform results in a much stronger decrease in/inhibition

of growth rates above and below optimum temperatures than is observed for any of the above equations. This feature was chosen by us as it is backed up by response data from multiple E. huxleyi strains in Zhang et al. 2014 Between- and within-population variations in thermal reaction norms of the coccolithophore Emiliania huxleyi Limnology and Oceanography, 59(5), 1570–1580.

-Page 7, line 4: Unneeded commas before and after "relatively simple"...or just rewrite for clarity "As such we wanted to examine how projections of productivity using our relatively simple equation compared to coccolithophorid productivity patterns observed in natural systems"

We will adopt this suggestion.

-Page 7, line 13: A citation of the CCPP metric is needed.

We will adopt this suggestion.

-Page 8, line 14: Need citation for the PIC:POC ratios used for E. huxelyi and G. oceanica

This has been corrected as detailed in the response to reviewer 1.

-Page 8, last paragraph: I took me awhile to figure out the CCPP estimates were made in three ways: 1) just E. huxleyi 2) just G. oceania 3) both species combined Is this correct? Only results for E. huxleyi CCPP was presented so maybe clarify here that only the results with the highest correlation to satellite PIC are shown. It's confusing because there are details described in the previous paragraphs about deriving CCPP for each species but then the results only show E. huxleyi CCPP.

Yes, the estimates were made using just E. huxleyi, just G. oceanica and then both species combined. Only results for E. huxleyi were presented as G. oceanica alone and in combination with E. huxleyi did not provide as good a correlation to satellite PIC. We shall mention at the end of this method section "While three CCPP scenarios are presented above, only the results with the highest correlation to satellite PIC are

shown and discussed below."

-Page 8, lines 26 and 27: Need parentheses around year for citations Gregg and Casey (2007) and Longhurst (2007).

We will adopt this suggestion.

-Results: -Page 9, Results section in general: Please specify in the headings that these are only the results for E. huxleyi (not G. oceanica).

We will adopt this suggestion (i.e. Change to "E. huxleyi responses to . . ..." for sections 3.1, 3.2 and 3.3).

-Page 9, line 2: Perhaps develop this small section a bit more. Which rate showed the best fit?

We will change the sentence to "The fit equation (Eq. 2) was able to explain up to 85% of growth, 80% of calcification and 73% of photosynthetic rate variability in E. huxleyi across a broad range of carbonate chemistry (25-4000 $\mu$atm), light (50-1200 $\mu$mol photons m-2s-1) and temperature (10-20oC) conditions (Table 1)."

-Page 9, line 6: Instead of just saying "all rates", please remind the reader what metabolic rates you are examining and refer to the equation presented in the methods.

We will change the sentence to "Based on fits of equation 2, growth, calcification and photosynthetic carbon fixation rates all had. . ... ."

-Page 9, line 7: It's hard to understand exactly what to look at in Table 2 and 3 to support this sentence (2nd sentence of the paragraph). It seems like CO2 concentrations of K1/2sat range form 0.85 to almost 5$\mu$mol kg-1 depending on light and temperature...

The difference in K1/2 sat concentration between treatments is not what is important here. Rather it is the difference in K1/2 sat between the different processes for the same conditions that supports this sentence. Under all conditions the difference in

CO2 concentration, between the three processes, required to support half of maximum rates is less than 1-2 $\mu$mol kg-1. We will clarify this issue in the revision.

-Page 9, lines 8-10: Mention what are the optimal CO2 concentrations and put this into units of $\mu$atm to make it more relatable to the reader. Are we at the optimum CO2 already for coccolithophores or will it come in the near future? At what CO2 concentrations is K1/2inhib reached? More specifics would give the reader more useful information.

We will add the CO2 concentrations for optima and K1/2 inhib. The reason we use CO2 concentrations rather than fugacities is that for the same concentrations, the fugacity would be different for two temperatures.

-Page 9, line 14/15: What columns in table 2 are the reader supposed to be looking at? Are you referring to the Vmax column?

Yes. The Vmax not only represents the maximum rate in a treatment, but also is where we see the greatest change in rates due to temperature and light. This is because Vmax is achieved under optimal CO2 conditions and, based on our findings, rates under optimal CO2 conditions are the ones which are most sensitive to changes in temperature and light conditions.

-Page 9, line 18: I had to read this sentence several times before I actually understood it. Would this be a better way to put this?: "CO2 half saturation concentration were insensitive to temperature. However, under increasing temperatures CO2 optima for growth and inhibition occurred at lower CO2 concentrations"

Yes, with some modification. Changed to "CO2 half saturation concentrations were insensitive to temperature (Table 2). However, under increasing temperatures CO2 concentrations for both optimal growth and for inhibition of rates to half the maximum (K1/2CO2 inhib) decreased (Table 2)."

-Discussion: -Page 10, line 6: Since this is a major conclusion of the paper, it should

be shown directly somehow. All the original G. oceanica data is published elsewhere, so a graphical summary of BOTH the E. hux and G. oceanic data would be helpful. This could be done through line plots comparing the metabolic rates of the two species under varying CO2 or in a bar plot comparing the rates. I just think it's necessary to show a visual comparison of E hux and G oceanica data (or data-derived function) since the title of the paper indicates a comparison.

Actually, the data for the response of G. oceanica to CO2 under different light conditions is already presented for easy comparison in a supplementary table. We will add this cross-reference into the paper. This table is already referenced multiple times in the paper and we do not wish to repeat information by also presenting it in graphic form.

The data for the response of G. oceanica to CO2 under different temperatures is the only data not available for direct comparison to E. huxleyi in this paper and we feel it does not add enough to this paper to be included here as well. Besides this the main focus of the comparison between the species for this paper is in the fundamental and realised niche descriptions.

-Page 10, line 30: A change in CO2 optima of $11\mu$mol kg-1 is not that small.

We will change the sentence to "Changes in temperature produced little (<1 $\mu$molkg-1) change in CO2 substrate half-saturation (K1/2CO2 sat) levels, at least within the measured range (Figure 1, Table 2). CO2 requirements for optimum rates had a tendency to slightly decrease with warming temperatures. Similar results were observed for. . . . . .. . . . . .."

-Page 11, line 5: Unneeded commas around "at least some"

We will change this.

-Page 11, line 15: Again, here is where a comparison figure between E. hux and G. oceanica would be helpful.

All information is available in the accompanying tables.

-Page 12, line 3: The range tested in this study is so much higher than even what projected under RCP8.5 at the end of the century. The temperature range tested in this study is much more realistic. How warm would the world be under $5000\mu$atm CO2?

Please see our response to the comment on Page 3.

-Page 12, line 12: I think a major limitation of this study is the focus on just one strain of one morphotype of E. huxelyi. Different E. huxelyi morphotypes show significant genetic and physiological variability (see Read et al., 2013; Langer et al., 2009; Krumhardt et al., 2017). Accounting for these differences could add significant uncertainty to the conclusions. I think that the last sentence of section 4.4 (before section 4.4.1) would fit better in a section on the "Limitations of this study" at the end of the Discussion section, where you describe how E huxleyi strain PML B92/11 is used to be representative of all E. huxleyi for determining niche and projections under future CO2 and warming in this study. This doesn't make the results invalid, but is just a limitation that needs to be made clearer. This would then lead in nicely with the conclusion that more testing with colder water strain/species/morphotypes of E huxleyi is necessary.

Please refer to our reply to the first comment of reviewer one for our response on the limitations of using a single strain. In terms of creating a limitations section, we believe it makes it easier to follow the paper, and remind readers of its limitations, if we mention the specific limitations of our research not as a separate section but rather as part of the discussion for each section. In this way it can be made more clear what the limitations are and what they mean for each section.

-Page 12, line 22: Capitalize "Figure"

We will change this.

-Page 13, line 7: I think it's well established that E. huxleyi is a generalist, given its widespread distribution from subpolar to tropics.
We will change to "indicating that this species is more of a generalist than G. oceanica".

-Page 13, line 9: Unneeded comma after "niche"

This will be removed.

-Page 13, line 25: Reference needed for this E. huxleyi warm water strain that outcompete G. oceanica at temps > 25C

This observation is based on the data compiled in figure 6. The data sources will be referenced in the figure caption.

-Page 13, line 32/33: I'm confused by this lower CO2 extreme of 25$\mu$atm. By Figure 5 it looks like G. oceanica outcompetes E hux at temps > 25C at 25$\mu$atm CO2.

Changed to "At extreme CO2 levels of 25 and 4000 $\mu$atm G. oceanica is only projected to reach higher growth rates than E. huxleyi at temperatures above 25.5 and 29oC, respectively (Figure 5)."

-Page 14, lines 1-3: This is a major finding of this study and should be put in the abstract.

It is already mentioned on line 6. Nevertheless, we could add "However, the greater sensitivity of G. oceanica to increasing [CO2] is partially mitigated by increasing temperatures."

-Page 14, line 4/5: The sentence seems like it shouldn't have the "under a broader range of CO2 conditions" part at the end. Under higher temperature alone (holding CO2 at about 400$\mu$atm) G. oceanica outcompetes E hux at temps > 22C. Or perhaps I'm misunderstanding this sentence completely?

Yes, this is a misunderstanding. What we mean by this is that as temperatures alone increase, the range of CO2 conditions under which G. oceanica outcompetes E. huxleyi becomes broader (i.e. expands from 300-500$\mu$atm to 200-600$\mu$atm). We will make this more clear within the section with the following change "Under increasing temperatures, but constant CO2 levels, the range of CO2 conditions under which G. oceanica outcompetes E. huxleyi expands (e.g. from ∼100-600$\mu$atm at 24oC to ∼250-1100$\mu$atm at 26oC)".

-Page 14, last paragraph of section 4.4.2: This would be better in a "Limitations of this study" section, as mentioned above.

See reply for the Page 12, line 12 comment.

-Page 15, line 12: By "productivity", do you mean calcification?

It is production of particulate inorganic carbon. This will be clarified.

-Page 15, lines 14-22: Would this paragraph better fit in the Results section?

We have opted to leave it where it is as it is part of the general discussion on how well our CCPP estimates fit to satellite derived CCPP.

-Page 16, lines 8-20: Could it be that E huxleyi CCPP just matches better because it's so much more abundant than G. oceanica?

While that would help, abundance alone does not completely control global PIC production. It is also the ratio of abundance/ growth to PIC production. The thought was that adding G. oceanica might help improve the fit by accounting for the greater amount of PIC production made by more heavily calcifying species in warmer regions.

-Page 16, line 14/15: I do not understand this sentence. So the combined CCPP in the North Pacific and Atlantic is greater or less than the E huxleyi CCPP?

Yes, as mentioned at the end of the sentence, all differences are relative to the E. huxleyi alone fit.

-Tables: -Tables 2 and 3: Put parentheses around units for K1/2CO2inhib and K1/2CO2sat in tables.

We will adopt this suggestion.

-Figures: -Figures 1 and 2: Indicate that this data is just for E. huxleyi in the caption. Also, show relevant CO2 range with a shaded area as in Sett et al., 2014 and indicate average oceanic CO2 concentration at present day.

We will adopt the first suggestion. As for the second, we will add a range of current day oceanic CO2 concentrations. Based on the carbonate chemistry data used for our global projections, modern CO2 concentrations range from 8.45-29.94 $\mu$mol kg-1. We shall add these boundaries as a shaded area in figures 1 and 2.

-Figure 3: Each "slice" looks the same.. maybe there's a better way to show differences between light levels or lack thereof? Also I do not understand the colors – add color legend.

Figure 3 is a full three dimensional niche comparison between E. huxleyi and G. oceanica. The visual similarity of slices at different light levels shows an important point, i.e. a small influence of light in modulating the CO2 and temperature response. A figure legend will be added.

-Figure 4: Make the $\mu$EH $>\mu$GO bigger or put it next to the color bar. It's a bit hard to notice and this is critical for understanding the figure.

We will make the font bigger.

-Figure 5: same suggestion as for Figure 4.

We shall adopt the requested changes.

-Figure 7: It needs to be mentioned in the caption that these maps are CCPP for Ehuxleyi only.

We will adopt the suggestion.

-Figure 8: Again, this is just CCPP for E huxleyi, right? This should be indicated in the figure caption. Also, a little map of the provinces (like in the supplemental section) would be great next to these bar plots. Having a map next to this data would make the

figure much more relatable.

Yes it is, and this will be made clearer. We have moved the map into the same figure as the bar plots.

---

## Author Response (AR1)

A.J. Poulton (Referee) • Comment 1. Is strain PML B92/11 (isolated from Bergen, Norway) the best strain of *E. huxleyi* to present a niche description of *E. huxleyi*? My query here is whether the authors have considered the potential need to consider multiple strains when trying to describe the fundamental and realised niche of the species *E. huxleyi*. Though the authors state that cold-water (Southern Ocean) strains need to be considered more, would not a broader study of several strains of *E. huxleyi*, isolated from various geographical regions, result in a better description of the species as a whole? Related to this is whether the authors have considered examining different (geographically) strains of *G. oceanica* and whether the limitations of *G. oceanica*'s niche could relate to the limited number of strains available for this species? These comments are not meant to detract from the present study, but rather emphasize the broader context.

We agree that considering multiple strains, from diverse ocean regions, would benefit our study in describing the fundamental and realised niches for a species in more general terms. Nevertheless, despite the fact that our realised niche projections are based on only one strain for each species, they do generally fit to modern day observations. This indicates that the differences in requirements and sensitivities of the two species as described here are large enough to be revealed by choosing only two representatives. This has now been detailed in the final paragraph of section 4.4.2 (page 14 lines 31-33 and page 15 lines 1-3) as follows "Considering multiple strains, from diverse ocean regions, would aid our study in describing the fundamental and realised niches for a species in more general terms. However, even though our realised niche projections are based on only one strain for each species, they do generally agree with experimental observations of other strains, and with planktonic and sediment observations of each species as a whole. This indicates that the differences in requirements and sensitivities of the two species as described here are large enough to be revealed by choosing only one representative for each species."

• Comment 2. The authors use a 'recently proposed metric' for coccolithophore calcification rates (CCPP), but proposed by who? No reference is mentioned in the paper. Could the authors provide more context and information on this new metric?

This metric was proposed by us in a recently published paper. We have added the reference for this metric on page 3 line 14.

Specific points • Pg 1, Ln 4 (Ln 29) – *Emiliania huxleyi* is certainly one of the most abundant species, but not sure if *G. oceanica* can be classified in the same category. The two are common, though *E. huxleyi* has such a broad bio-geographical range compared to a narrower one for *G. oceanica* and generally a tropical range. Maybe relative abundance is not the characteristic to emphasize and either a commonality in many coccolithophore communities or bloom-formation by the two is more relevant (to global PIC production).

We agree and have changed the wording on page 1 lines 4-5 to: "the two most common bloom-forming species in present day coccolithophore communities appear adapted to..."

• Pg 1, Ln 13 – As well as the $R^2$ of the correlation, it would be good to know what the slope of the line looks like and the p-value in the abstract.

We have added the p-value and slopes to the abstract and discuss the slope of the line on page 16 lines 10-20.

• Pg 8, Ln 13-15 – Have the authors considered how total cell carbon (PIC+POC) to PON ratios would influence their data? In many ways, the N requirement of a coccolithophore cell is to produce both the PIC and POC. Also, are the PIC:POC ratios of 1 and 2 for *E. huxleyi* and *G. oceanica*, respectively, averages of the values given on Lines 23-24? Some justification for the use of the these values, given the ranges known in the literature, is needed.

We have justified/explained total cell carbon:PON and the PIC:POC values as follows. "We first assumed a Redfieldian ratio of 106:16 C:N to determine the maximum POC production possible from the amount of available nitrate. We then calculated the amount of PIC which would be co-produced based on a mean PIC:POC. The average PIC:POC of *E. huxleyi* and *G. oceanica* was calculated as the average of all treatments between 300-1000 µatm from Sett et al. (2014), Zhang et al. (2015) and this study. Based on these averages (PIC:POC of 0.8 and 1.35 for *E. huxleyi* and *G. oceanica*, respectively), and assuming Redfieldian production a corresponding PIC:PON of 5.3 and 8.94 was calculated." This explanation has now been included on page 8 (lines 31-33) and 9 (lines 1-3).

• Pg 14, Lns 21-23 – Surprised the review article by Monteiro et al. (2016) is not mentioned when considering viral attack and top-down effects as this article concluded that these were key considerations in the ecology of coccolithophores.

We have added a reference to the review article on pge 15 lines 16-17. • Pg 16, Lns 2 and 3 – Rather than citing the PhD thesis of Charalampopoulou (2011), why don't the authors cite the peer-reviewed papers derived from this piece of work that address these points? Charalampopoulou et al. (2011) Irradiance and pH affect coccolithophore community composition on a transect between the North Sea and the Arctic Ocean. Marine Ecology Progress Series 431, 25-43, doi: 10.3354/meps09140. Charalampopoulou et al. (2016) Environmental drivers of coccolithophore abundance and calcification across Drake Passage

(Southern Ocean). Biogeosciences 13, 5917-5935, doi: 10.5194/bg-13-5917-2016.

We have adopted the reviewer's suggestion on page 16 lines 34 and 35.

• Pg 16, Ln 23 – Consider the use of the term 'benefit' in terms of *E. huxleyi* outcompeting *G. oceanica*.

Instead we have stated on page 17 line 20 "*E. huxleyi* will gain further competitive advantage over G. oceanica."

• Pg 16, Ln 27 – What are coccolithophore dominated ecosystems? Please phrase in a more specific way (e.g. where coccolithophores are abundant enough to potentially influence the air-sea $CO_2$ flux (e.g. coccolithophore blooms) or dominate the deep-sea flux of particulate material (e.g. subtropical gyres). Coccolithophores never dominate ecosystems.

We have changed this on page 17 lines 22-25 to: "Such changes could have significant implications for climate feedback mechanisms, one being the relative strengths of the organic and inorganic carbon pumps in ecosystems where coccolithophores are abundant enough to significantly impact the air-sea $CO_2$ flux (e.g. coccolithophore blooms) and/or dominate the deep-sea flux of particulate material (e.g. subtropical gyres)."

Figures. • Fig 3 - Missing legend that is on Fig 4, consider swapping figures around or reproducing the legend.

A legend has been added to Figure 3.

• Fig 6 – This appears to be a rather unconvincing relationship. Is it possible to plot the 95% CI limits for the relationship? In addition, is there a sampling depth issue here that results in greater amount of data at high temperatures? That is to say, is the distribution of data related to more shallow tropical sediment samples than deep cold sub-polar sediment samples?

We have added 95% prediction bounds based on new observations for the global relationship. The fact that only the Atlantic basin does not entirely follow the trend has been mentioned in the text of the paper in more detail now. On page 14 lines 17-20 we have stated the following: "It is noted, however, that samples from the south-equatorial to equatorial Atlantic Ocean in Boeckel et al. (2006) do not follow the general temperature trend observed in other ocean basins (Figure 6). In this location it appears that *G. oceanica* abundance is driven more by increasing nutrient concentrations than by temperature. It seems oceanic upwelling in this region is driving a different relationship between *E. huxleyi* and *G. oceanica* than observed in other areas." There is very little sediment sampling data at high latitudes in general let alone which contains these species. There is a disproportionately large amount of sampling in the warmer tropical areas. Also, we selected only samples which were above the lysocline and therefore were not affected by the possible confounding effects of differential dissolution of coccoliths.

• Fig 8 – Would it be more appropriate to plot as scatter plots where each data point is from each province? Maybe this would emphasize better how well the two agree and in which provinces they do not agree?

While using a scatterplot does emphasise that the two do not agree in some provinces, it also makes it more difficult to determine which provinces do and do not agree. For the purpose of clearly comparing each province in each season quickly, and having now included more details and a discussion on the slope of the fits, we feel that the barplot works best.

Anonymous (Referee 2) Abstract: • As I understand it, the inhibitory effect of increasing $CO_2$ on *G. oceanica* is the main reason for this species' projected contraction under a future scenario. This should be emphasized in the abstract (1). As it is now, the projection of a contracted *G. oceanica* niche is surprising because it is generally the warmer water adapted species. Also, since *E. huxleyi* CCPP shows a better correlation with satellite-derived PIC than when combine with *G. oceanica*, this should be mentioned in the abstract (2). Otherwise, given the title of the paper, one assumes that the CCPP estimates are derived from partitioning niches between *E. huxleyi* and *G. oceanica*. Also, maybe a sentence at the beginning of Abstract describing why these two particular species are being compared would be helpful.

1. We have modified page 1 lines 9-10 to "For a future RCP 8.5 climate change scenario (1000 μatm fCO$_2$) we project a $CO_2$ driven niche contraction for *G. oceanica* to regions of even higher temperatures." 2. We have modified page 1 lines 13-16 to "Based on *E. huxleyi* alone, as there was interestingly a better correlation than when in combination with *G. oceanica*, and excluding the Antarctic province from the analysis we found a good correlation between CCPP and satellite derived PIC in the other regions 15 with an $R^2$ of 0.73, p<0.01 and a slope of 1.03 for Austral winter/Boreal summer and an $R^2$ of 0.85, p<0.01 and a slope of 0.32 for Austral summer/Boreal winter." 3. We have modified lines 4-5 of the abstract to "Based on our analysis, the two most common bloom-forming species in present day coccolithophore communities appear to be……."

Intro: • Page 2, lines 3-9: This paragraph on future changes to the surface ocean environment needs expanding. What happens to nutrient availability with increasing stratification? How could this affect $CaCO_3$ production and growth rate in coccolithophores? How could increasing $CO_2$ affect growth rate/ calcification of coccolithophores? The impact of increasing light is described but not the other effects of climate change. Increasing temperature would also increase metabolic rates, unless

nutrient limitation becomes too strong. Overall this paragraph just needs more development with respect to the effects of anthropogenic climate change on coccolithophore habitat and how each effect could impact growth/calcification.

The potential effects of $CO_2$ on phytoplankton in general and coccolithophores in particular are already covered in the previous and following paragraph. Nevertheless, we have modified the text on page 2 lines 4-11 to "Under current scenarios, ocean temperatures are projected to increase from 2.6 to 4.8°C by 2100 (IPCC, 2013b). In addition, warming of the ocean is expected to enhance vertical stratification of the water column, resulting in a shoaling of the surface mixed layer and increasing overall light and decreasing nutrient availability in the euphotic zone (Bopp et al., 2001; Rost and Riebesell, 2004; Lefebvre et al., 2012). While increased light intensity and temperatures often accelerate growth in phytoplankton, excessive levels of light and temperature can cause damage to the photosynthetic apparatus and reduce effectiveness of enzymes thus decreasing growth (Powles, 1984; Rhodes et al., 1995; Crafts-Brandner and Salvucci, 2000; Zondervan et al., 2002; Helm et al., 2007; Pörtner and Farrell, 2008). Meanwhile, reduced nutrient availability could diminish overall productivity."

• Page 2 line 18: There needs to be a paragraph with some background about the two species discussed in this paper. Why are you comparing these two particular coccolithophore species? These are the two major bloom forming coccolithophores. It is well known that *E. huxleyi* is very widespread, but how abundant is *G. oceanica*? Where does *G. oceanica* tend to thrive? Also mention that there are several different mophotypes of *E. huxleyi* and how they might differ. A bit of biogeography background would be helpful. This would then lead into the fundamental vs. realized niche paragraph.

We have added the following text in page 2 line 15 "The coccolithophores *Emiliania huxleyi* and *Gephyrocapsa oceanica* are considered the most common species in present day coccolithophore communities. *E. huxleyi* is a ubiquitous coccolithophore species having been observed from polar to equatorial regions, nutrient poor ocean gyres to nutrient rich upwelling systems, and from the bright sea surface down to 200m depth (McIntyre Be 1967; Winter et al. 1994; Hagino Okada 2006; Boeckel Baumann 2008; Mohan et al. 2008; Henderiks et al. 2012). The wide tolerance of *E. huxleyi* to different environmental conditions is believed to be, at least partially, explained by the existence of several environmentally selected ecotypes and morphotypes within the species (Paasche 2001; Cook et al. 2011). *G. oceanica* is also found in most oceanographic regions (McIntyre and Be 1967; Okada and Honjo 1975; Roth and Coulbourn 1982; Knappertsbusch et al. 1993; Eynaud et al., 1999; Andruleit et al. 2003; Saaveda-Pellitero et al. 2010), however with a tendency towards warmer waters with very few specimens observed below 13oC (McIntyre and Bé, 1967; Eynaud et al., 1999; Hagino et al., 2005)."

• Page 2 lines 28-35: the CCPP- PIC comparison is left out of this paragraph. It would be good to mention this here to indicate how it ties in with the *E. hux – G. oceanica* niche comparison.

We have added the following to the end of the paragraph on page 3 lines 5-14 "Finally, we compare satellite derived particulate inorganic carbon estimates with a recently proposed metric for coccolithophore success on the community level, i.e. the temperature, light and carbonate chemistry speciation dependent calcium carbonate potential (Gafar et al. 2018)."

Methods: • Page 3, line 4: Why test such high $CO_2$ values? Are these even realistic? For instance, if end of the century $CO_2$ concentration of 985atm (about 50mol kg$^{-1}$ aqueous $CO_2$), corresponds to a 4.8°C temperature increase, then why go up to 250mol kg$^{-1}$ $CO_2$? The range of $CO_2$ is therefore bigger than the temperature range in terms of real world conditions. An explanation for this experimental setup would be helpful.

Fitting non-linear responses of multiple stressors to data requires a broad range of environmental conditions, as otherwise the shaping factors of limitation and inhibition are lost (absent from data while present in model equation). With this broader range we also have the added benefit for identifying tipping points and changes in sensitivities to $CO_2$ with changing light and temperature. We have made our rational more clear in the methods section by adding the following to page 3 line 17: "To accurately identify optimal conditions, tipping points and sensitivities of rates in response to changing $CO_2$, light and temperature, a broad range of experimental conditions are required. Hence, mono-specific……."

• Page 3, section 2.1: The authors need to mention the particular *E. hux* morphotype being tested (PML B92/11 is morphotype A).

We have added the requested information to page 3 line 19.

• Page 4, line 22: Why would there be a lag phase? It seems the growth rate is calculated correctly (after the lag phase is over), but a quick explanation of why there is a lag phase at extreme $CO_2$ and whether this is a normal phenomenon in phytoplankton culturing and physiological testing would be helpful.

We have added the following information to page 5 lines 4-8. "At both, the extreme low and high $CO_2$ treatments, carbonate chemistry at the end of the pre-incubation phase can significantly deviate from initial and hence experimental treatment conditions due to enhanced air/water $CO_2$ gas exchange during regular cell abundance monitoring. As a result, at some extreme $CO_2$ levels there was an initial lag phase and therefore growth rates were calculated from densities only during the exponential part of the growth phase."

• Page 5, section 2.7: I find this section about the data transformation confusing, particularly about the temperature. Is this just for growth rate? How do the resulting temperature-dependent growth rates compare to other studies on coccolithophores (Fielding 2013, Buitenhuis et al., 2008)?

This transform is applied to all rates to reduce skew and are common practice in multi-variate fitting procedures. As mentioned within section 2.7, this temperature transform compares well to other temperature dependant growth rate equations such as the single species responses to the Eppley temperature envelope curve and the Norberg model. Our temperature-dependant growth rate estimates show a similar response to the optimal growth function in Buitenhuis et al. 2008 and the Flinn equation in Fielding et al. 2013. The power function in Fielding et al. 2013 also follows a similar pattern, of growth rate increase with rising temperature, as our transform but lacks a term to inhibit rates as temperatures rise above optimum. However, our temperature transform results in a much stronger decrease in inhibition of growth rates above and below optimum temperatures than is observed for any of the above equations. This feature was chosen by us as it is backed up by response data from multiple *E. huxleyi* strains in Zhang et al. 2014 Between- and within-population variations in thermal reaction norms of the coccolithophore *Emiliania huxleyi* Limnology and Oceanography, 59(5), 1570–1580.

• Page 7, line 4: Unneeded commas before and after "relatively simple"...or just rewrite for clarity "As such we wanted to examine how projections of productivity using our relatively simple equation compared to coccolithophorid productivity patterns observed in natural systems"

We have rewritten the sentence as suggested above on page 7 lines 21-22.

• Page 7, line 13: A citation of the CCPP metric is needed.

The citation for the CCPP metric is now referenced on page 7 line 32.

• Page 8, line 14: Need citation for the PIC:POC ratios used for *E. huxelyi* and *G. oceanica*

This has been corrected as detailed in the response to reviewer 1.

• Page 8, last paragraph: I took me awhile to figure out the CCPP estimates were made in three ways: 1) just *E. huxleyi* 2) just *G. oceanica* 3) both species combined Is this correct? Only results for *E. huxleyi* CCPP was presented so maybe clarify here that only the results with the highest correlation to satellite PIC are shown. It's confusing because there are details described in the previous paragraphs about deriving CCPP for each species but then the results only show *E. huxleyi* CCPP.

Yes, the estimates were made using just *E. huxleyi*, just *G. oceanica* and then both species combined. Only results for *E. huxleyi* were presented as *G. oceanica* alone and in combination with *E. huxleyi* did not provide as good a correlation to satellite PIC. We have now stated on page 9 lines 19-20 "While three CCPP scenarios are presented above, only the results with the highest correlation to satellite PIC is shown and discussed below."

• Page 8, lines 26 and 27: Need parentheses around year for citations Gregg and Casey (2007) and Longhurst (2007).

We have adopted this suggestion on page 9 lines 14 and 15.

Results: • Page 9, Results section in general: Please specify in the headings that these are only the results for *E. huxleyi* (not *G. oceanica*).

We have adopted this suggestion by changing headings for sections 3.1, 3.2 and 3.3 to "*E. huxleyi* responses to . . . .".

• Page 9, line 2: Perhaps develop this small section a bit more. Which rate showed the best fit?

We have changed the sentence at the start of section 3 (page 9 lines 22-23) to "The fit equation (Eq. 2) was able to explain up to 85% of growth, 80% of calcification and 73% of photosynthetic rate variability in *E. huxleyi* across a broad range of carbonate chemistry (25-4000 µatm), light (50-1200 µmol photons m$^{-2}$s$^{-1}$)) and temperature (10-20°C) conditions (Table 1)."

• Page 9, line 6: Instead of just saying "all rates", please remind the reader what metabolic rates you are examining and refer to the equation presented in the methods.

We have changed the sentence at the start of section 3.1 (page 9 line 26) to "Based on fits of Eq. (2), growth, calcification and photosynthetic carbon fixation rates all had. . . . . . "

• Page 9, line 7: It's hard to understand exactly what to look at in Table 2 and 3 to support this sentence (2nd sentence of the paragraph). It seems like $CO_2$ concentrations of K1/2 sat range form 0.85 to almost 5 mol kg$^{-1}$) depending on light and temperature...

The difference in K1/2 sat concentration between treatments is not what is important here. Rather it is the difference in K1/2

sat between the different processes for the same conditions that supports this sentence. Under all conditions the difference in $CO_2$ concentration, between the three processes, required to support half of maximum rates is less than 1-2 µmol kg$^{-1}$. We have clarified this by changing this sentence (page 9 lines 27-29) to "Growth, calcification and photosynthetic carbon fixation rates required similar $CO_2$ concentrations, with differences of less than 3 µmol kg$^{-1}$ under comparable temperature and light conditions, to stimulate rates to half the maximum, K1/2CO$_2$sat (Table 2, Table 3).".

• Page 9, lines 8-10: Mention what are the optimal $CO_2$ concentrations and put this into units of atm to make it more relatable to the reader. Are we at the optimum $CO_2$ already for coccolithophores or will it come in the near future? At what $CO_2$ concentrations is K1/2 inhib reached? More specifics would give the reader more useful information.

We have added the $CO_2$] concentrations for optima and K1/2 inhib on page 9 lines 30-31 and page 10 lines 1-2. The reason we use $CO_2$ concentrations rather than fugacities is that for the same concentrations, the fugacity would be different for two temperatures.

• Page 9, line 14/15: What columns in table 2 are the reader supposed to be looking at? Are you referring to the Vmax column? Yes. The Vmax not only represents the maximum rate in a treatment, but also is where we see the greatest change in rates due to temperature and light. This is because Vmax is achieved under optimal $CO_2$ conditions and, based on our findings, rates under optimal $CO_2$ conditions are the ones which are most sensitive to changes in temperature and light conditions.

• Page 9, line 18: I had to read this sentence several times before I actually understood it. Would this be a better way to put this?: "$CO_2$ half saturation concentration were insensitive to temperature. However, under increasing temperatures $CO_2$ optima for growth and inhibition occurred at lower $CO_2$ concentrations"

We have changed these sentences (page 10 lines 6-8) to "$CO_2$ half saturation concentrations (K1/2CO$_2$sat) were insensitive to temperature (Table 2). However, under increasing temperatures $CO_2$ concentrations for both optimal growth and for inhibition of rates to half the maximum (K1/2CO$_2$inhib) decreased (Table 2)."

Discussion: • Page 10, line 6: Since this is a major conclusion of the paper, it should be shown directly somehow. All the original *G. oceanica* data is published elsewhere, so a graphical summary of BOTH the *E. hux* and *G. oceanica* data would be helpful. This could be done through line plots comparing the metabolic rates of the two species under varying $CO_2$ or in a bar plot comparing the rates. I just think it's necessary to show a visual comparison of *E hux* and *G. oceanica* data (or data-derived function) since the title of the paper indicates a comparison.

Actually, the data for the response of *G. oceanica* to $CO_2$ under different light conditions is already presented for easy comparison in a supplementary table. We have added this cross-reference to page 11 line 9. This table is already referenced multiple times in the paper and we do not wish to repeat information by also presenting it in graphic form. The data for the response of *G. oceanica* to $CO_2$ under different temperatures is the only data not available for direct comparison to *E. huxleyi* in this paper and we feel it does not add enough to this paper to be included here as well. Besides this, the main focus of the comparison between the species for this paper is in the fundamental and realised niche descriptions.

• Page 10, line 30: A change in $CO_2$ optima of 11 mol kg$^{-1}$ is not that small.

We have changed this (page 11 lines 19-21) to "Changes in temperature produced little (<1 µmolkg$^{-1}$) change in $CO_2$ substrate half-saturation (K1/2CO$_2$sat) levels, at least within the measured range (Figure 1, Table 2). $CO_2$ requirements for optimum rates tended to slightly decrease with warming temperatures. Similar results were observed for............."

• Page 11, line 5: Unneeded commas around "at least some"

Commas on page 11 line 29 have been removed.

• Page 11, line 15: Again, here is where a comparison figure between *E. hux* and *G. oceanica* would be helpful.

All information is available in the accompanying tables.

• Page 12, line 3: The range tested in this study is so much higher than even what projected under RCP8.5 at the end of the century. The temperature range tested in this study is much more realistic. How warm would the world be under 5000atm $CO_2$? Please see our response to the comment on Page 3 line 4.

• Page 12, line 12: I think a major limitation of this study is the focus on just one strain of one morphotype of *E. huxleyi*. Different *E. huxelyi* morphotypes show significant genetic and physiological variability (see Read et al., 2013; Langer et al., 2009; Krumhardt et al., 2017). Accounting for these differences could add significant uncertainty to the conclusions. I think that the last sentence of section 4.4 (before section 4.4.1) would fit better in a section on the "Limitations of this study" at the end of the Discussion section, where you describe how *E. huxleyi* strain PML B92/11 is used to be representative of all *E. huxleyi* for determining niche and projections under future $CO_2$ and warming in this study. This doesn't make the results

invalid, but is just a limitation that needs to be made clearer. This would then lead in nicely with the conclusion that more testing with colder water strain/species/morphotypes of *E. huxleyi* is necessary.

Please refer to our reply to the first comment of reviewer one for our response on the limitations of using a single strain. In terms of creating a limitations section, we believe it makes it easier to follow the paper, and remind readers of its limitations, if we mention the specific limitations of our research not as a separate section but rather as part of the discussion for each section. In this way it can be made more clear what the limitations are and what they mean for each section.

• Page 12, line 22: Capitalize "Figure"

The word figure has been capitalised (page 13 line 13).

• Page 13, line 7: I think it's well established that *E. huxleyi* is a generalist, given its widespread distribution from subpolar to tropics.

We have changed this (page 13 lines 30-31) to "For example, *E. huxleyi* is projected to reach higher growth rates than G. oceanica under a broader range of temperature, light and $CO_2$ conditions (Figures 3, 4 and 5), supporting the notion that *E. huxleyi* is rather a generalist.".

• Page 13, line 9: Unneeded comma after "niche"

Comma on page 14 line 1 has been removed.

• Page 13, line 25: Reference needed for this *E. huxleyi* warm water strain that outcompete *G. oceanica* at temps > 25°C

This observation is based on the data compiled in Figure 6. The data sources have been referenced in the figure caption. We have also re-worded the sentence on page 14 lines 20-21 to read "Globally the data suggests that dominance switches from *E. huxleyi* to *G. oceanica* at temperatures above 25°C which is similar to our projections."

• Page 13, line 32/33: I'm confused by this lower $CO_2$ extreme of 25 atm. By Figure 5 it looks like *G. oceanica* outcompetes *E hux* at temps > 25°C at 25 atm $CO_2$.

For clarity we have changed this section (page 14 lines 25-30) to read "$CO_2$ level also influences the relative growth rates of *E. huxleyi* and *G. oceanica*. Under current day levels of 400 µatm, *E. huxleyi* would dominate at temperatures up to 22°C (Figure 5). However, at higher and lower $CO_2$ levels, *E. huxleyi* begins to outgrow *G. oceanica* at progressively higher temperatures. At the same time, combined warming in a future ocean would partially mitigate the higher $CO_2$ sensitivity of *G. oceanica* (Figure 5). Nevertheless, over the naturally observed temperature range, *G. oceanica*'s niche would be expected to decrease towards higher $CO_2$ levels.".

• Page 14, lines 1-3: This is a major finding of this study and should be put in the abstract.

It is already mentioned on line 6. Nevertheless, we have added "However, the greater sensitivity of *G. oceanica* to increasing $[CO_2]$ is partially mitigated by increasing temperatures.".

• Page 14, line 4/5: The sentence seems like it shouldn't have the "under a broader range of $CO_2$ conditions" part at the end. Under higher temperature alone (holding $CO_2$ at about 400 atm) *G. oceanica* outcompetes *E hux* at temps > 22°C. Or perhaps I'm misunderstanding this sentence completely?

Yes, this is a misunderstanding. What we mean by this is that as temperatures alone increase, the range of $CO_2$ conditions under which *G. oceanica* outcompetes *E. huxleyi* becomes broader (i.e. expands from 300-500atm to 200-600atm). More importantly *G. oceanica* becomes less sensitive to high $CO_2$ conditions under elevated temperatures which is the point we wanted to highlight. We have made this more clear on page 14 lines 28-30 with the following change "At the same time, combined warming in a future ocean would partially mitigate the higher $CO_2$ sensitivity of *G. oceanica* (Figure 5). Nevertheless, over the naturally observed temperature range, *G. oceanica*'s niche would be expected to decrease towards higher $CO_2$ levels.".

• Page 14, last paragraph of section 4.4.2: This would be better in a "Limitations of this study" section, as mentioned above.

See reply for the Page 12, line 12 comment.

• Page 15, line 12: By "productivity", do you mean calcification?

It is production of particulate inorganic carbon (PIC). This has been clarified by changing the sentence to "The fact that three other global estimates, based on different sets of environmental parameters, all estimate very little PIC productivity in the Southern Ocean seems to support this theory." On page 16 lines 5-6.

• Page 15, lines 14-22: Would this paragraph better fit in the Results section?

We have opted to leave it where it is (page 16 lines 10-19) as it is part of the general discussion on how well our CCPP estimates fit to satellite derived CCPP.

• Page 16, lines 8-20: Could it be that *E. huxleyi* CCPP just matches better because it's so much more abundant than *G. ocean-*

*ica*?

While that would help, abundance alone does not completely control global PIC production. It is also the ratio of abundance/growth to PIC production. The thought was that adding *G. oceanica* might help improve the fit by accounting for the greater amount of PIC production made by more heavily calcifying species in warmer regions (Now on page 17 lines 5-17).

• Page 16, line 14/15: I do not understand this sentence. So, the combined CCPP in the North Pacific and Atlantic is greater or less than the *E. huxleyi* CCPP?

Yes, as mentioned at the end of the sentence, all differences are relative to the *E. huxleyi* alone fit (Now on page 17 lines 11-12).

Tables: • Tables 2 and 3: Put parentheses around units for K1/2$CO_2$inhib and K1/2$CO_2$sat in tables.

We have adopted this suggestion.

Figures: • Figures 1 and 2: Indicate that this data is just for *E. huxleyi* in the caption. Also, show relevant $CO_2$ range with a shaded area as in Sett et al., 2014 and indicate average oceanic $CO_2$ concentration at present day.

We have adopted the first suggestion by now stating "(A) Fitted particulate inorganic carbon (PIC), (B) particulate organic carbon (POC) production, and (C) growth rates (solid lines) of *E. huxleyi* in response....." in the caption. As for the second, based on the carbonate chemistry data used for our global projections, modern $CO_2$ concentrations range from 8.45-29.94 µmol kg$^{-1}$. We have added these boundaries as a shaded area in Figures 1 and 2.

• Figure 3: Each "slice" looks the same.. maybe there's a better way to show differences between light levels or lack thereof? Also I do not understand the colors – add color legend.

Figure 3 is a full three dimensional niche comparison between *E. huxleyi* and *G. oceanica*. The visual similarity of slices at different light levels shows an important point, i.e. a small influence of light in modulating the $CO_2$ and temperature response. A figure legend has now been added.

• Figure 4: Make the EH >GO bigger or put it next to the color bar. It's a bit hard to notice and this is critical for understanding the figure.

We have increased the font size.

• Figure 5: same suggestion as for Figure 4.

We have adopted the requested changes.

• Figure 7: It needs to be mentioned in the caption that these maps are CCPP for *E. huxleyi* only.

The caption has been amended to "Austral summer/Boreal winter (A) and Austral winter/Boreal summer (D) satellite measured particulate inorganic carbon. Austral summer/Boreal winter (B) and Austral winter/Boreal summer (E) *E. huxleyi* based CCPP estimates accounting for carbonate chemistry (substrate and hydrogen ion concentrations), light intensity and temperature.......".

• Figure 8: Again, this is just CCPP for *E. huxleyi*, right? This should be indicated in the figure caption. Also, a little map of the provinces (like in the supplemental section) would be great next to these bar plots. Having a map next to this data would make the figure much more relatable.

Yes, it is. We have moved the map into the same figure as the bar plots and changed the caption to "Satellite derived particulate inorganic carbon (black bars) and *E. huxleyi* based CCPP (white bars) estimates for major ocean biogeographical provinces as percentages of total production in (A) Austral winter/Boreal summer and (B) Austral summer/Boreal winter. (C) Major ocean biogeographical province definitions.".

Community comments Mario Cachao • You present a very interesting and useful peace of work. You selected the two species you refer as the most common. *Emiliania huxleyi* (Eh) is unquestionably the currently dominating species in oceanic niches. *Gephyrocapsa oceanica* (Go) is for sure the most abundant but in neritic domain (at least in my area, not sure about Australia), not exactly the most common in the overall oceans. In addition, from a paleoecological point of view, records of Eh are always compared to another small placolith species (small Gephyrocapsids; sG), not to Go, both in terms of relative and absolute abundances. I understand that Eh and Go are among those coccolithophores that better perform in cultures but shouldn't we compare Eh against sG instead? What's your opinion?

It is more that these two species are the most common in terms of their presence in coccolithophore communities rather than their dominance. Both species have a broad distribution across multiple ocean basins, for detail please see our response to reviewer 2 Page 2 line 18. It is this reason, plus the fact that data on responses to changing $CO_2$, temperature and light are available for both species, that we decided to compare the two species.

It would also be of interest to compare *E. huxleyi* against the small Gephyrocapsids. However, from what we understand the small Gephyrocapsids consist of multiple small Gephyrocapsa spp. which are not always identified to the species level (e.g. Table 3 Flores et al. 1999). As such, a niche comparison with *E. huxleyi* would be very difficult to accomplish from an experimental point of view.

30 *G. oceanica* is often mentioned alongside *E. huxleyi* in sediment core data (i.e. McIntyre and Be 1967, Chen and Shieh 1982, Roth and Coulburn 1982, Knappertsbusch et al. 1993, Findlay and Flores 2000, Andruleit and Rogalla 2002, Boeckel et al. 2006, Fernando et al. 2007, Saaveda-Pellitero et al. 2010). Further, it seems that in longer geological records that *E. huxleyi* is usually compared to larger Gephyrocapsa species such as *G. mullerae*, *G. caribbeanica* and *G. oceanica* as well as the small Gephyrocapsids (Flores et al. 1997, Findlay and Florin 2000, Flores et al. 2003, Backman et al. 2009). So, we believe it is
35 equally reasonable to compare *E. huxleyi* and *G. oceanica* as it is to compare *E. huxleyi* to the small Gephyrocapsids.

[revised manuscript text omitted]

---

## Referee Report (RR1)

**Second round of comments**

This manuscript is much improved. I have just a few minor comments:

Page 1, line 13: put a comma before and after 'i.e.'

Page 1, line 15: put a comma after 'analysis' and remove 'in other regions'

Page 13, lines 25-30: this paragraph really helps to put your findings into context--Interesting!